# Distinct mechanisms define murine B cell lineage immunoglobulin heavy chain (IgH) repertoires

Yang Yang[1][*][†], Chunlin Wang[2][†], Qunying Yang[2], Aaron B Kantor[1], Hiutung Chu[3], Eliver EB Ghosn[1], Guang Qin[1], Sarkis K Mazmanian[3], Jian Han[2][*], Leonore A Herzenberg[1][*]

[1]Genetics Department, Stanford University, Stanford, United States; [2]HudsonAlpha Institute for Biotechnology, Huntsville, United States; [3]Biology and Biological Engineering Department, California Institute of Technology, Pasadena, United States

**Abstract** Processes that define immunoglobulin repertoires are commonly presumed to be the same for all murine B cells. However, studies here that couple high-dimensional FACS sorting with large-scale quantitative IgH deep-sequencing demonstrate that B-1a IgH repertoire differs dramatically from the follicular and marginal zone B cells repertoires and is defined by distinct mechanisms. We track B-1a cells from their early appearance in neonatal spleen to their long-term residence in adult peritoneum and spleen. We show that de novo B-1a IgH rearrangement mainly occurs during the first few weeks of life, after which their repertoire continues to evolve profoundly, including convergent selection of certain V(D)J rearrangements encoding specific CDR3 peptides in all adults and progressive introduction of hypermutation and class-switching as animals age. This V(D)J selection and AID-mediated diversification operate comparably in germ-free and conventional mice, indicating these unique B-1a repertoire-defining mechanisms are driven by antigens that are not derived from microbiota.

*For correspondence: yang71@ stanford.edu (YY); jhan@ irepertoire.com (JH); leeherz@ stanford.edu (LAH)

[†]These authors contributed equally to this work

## Introduction

Follicular B (FOB), marginal zone B (MZB) and B-1a cells are the major mature B cell populations in the mouse. Although these B cell subsets all produce functionally important antibodies, they differ profoundly in function and developmental origin (*Kantor and Herzenberg, 1993*; *Hardy and Hayakawa, 2001*; *Baumgarth, 2011*). Previous studies have shown that B-1a cells are efficiently generated during fetal and neonatal life, and are maintained by self-replenishment in adult animals (*Hayakawa et al., 1985*; *Montecino-Rodriguez et al., 2006*; *Kantor et al., 1992*). In contrast, both FOB and MZB populations emerge later and are replenished throughout life by *de novo* development from bone marrow (BM) hematopoietic stem cells (HSC). Our recent studies show that BM HSC reconstitute FOB and MZB, but fail to reconstitute B-1a cells (*Ghosn et al., 2012*), which are derived from distinct progenitors at embryonic day 9 yolk sac (*Yoshimoto et al., 2011*).

For each B cell subset, their antibody responses are enabled by the basic processes that generate the immunoglobulin (Ig) structure. Multiple mechanisms contribute to creating the primary Ig heavy (IgH) and light chain (IgL) diversity. For IgH, these include combinatorial assortment of individual variable (V), diversity (D) and joining (J) gene segments, nucleotide(s) trimming in the D-J and V-DJ joining site, and, template-dependent (P-addition) and independent (N-addition) nucleotide(s) insertion at the joined junctions (*Yancopoulos and Alt, 1986*; *Kirkham and Schroeder, 1994*). The V(D)J joining processes define the third IgH complementarity-determining region (CDR3), which often lies at

**eLife digest** Our immune system protects us by recognizing and destroying invading viruses, bacteria and other microbes. B cells are immune cells that produce protective proteins called antibodies to stop infections. These cells are activated by 'antigens', which are fragments of molecules from the microbes or from our own cells. When an antigen binds to a B cell, the cell matures, multiplies and produces proteins called antibodies. These antibodies can bind to the antigen, which marks the microbe for attack and removal by other cells in the immune system.

Each antibody consists of two 'heavy chain' and two 'light chain' proteins. B cells are able to produce a large variety of different antibodies due to the rearrangement of the gene segments that encode the heavy and light chains. In mice, there are two kinds of B cells – known as B-1a and B-2 cells – that play different roles in immune responses. B-1a cells have long been known to produce the 'natural' antibodies that are present in the blood prior to an infection. On the other hand, B-2 cells produce antibodies that are specifically stimulated by an infection and are better adapted to fighting it. Previous studies have shown that both types of antibodies are required to allow animals to successfully fight the flu virus.

Here, Yang, Wang et al. used a technique called fluorescence-activated cell sorting (or FACS) and carried out extensive genomic sequencing to study how the B-1a and B-2 populations rearrange their genes to produce heavy chains. This approach made it possible to separate the different types of B cells and then sequence the gene for the heavy chain within the individual cells. The experiments show that the "repertoire" of heavy chains in the antibodies of the B-1a cells is much less random and more repetitive than that of B-2 populations.

Furthermore, Yang, Wang et al. show that B-1a cells produce and maintain their repertoire of heavy chains in a different way to other B-2 populations. B-1a cells develop earlier and the major genetic rearrangements in the gene that encodes the heavy chain occur within the first few weeks of life. Although the gene rearrangements have mostly stopped by adulthood, the B-1a antibody repertoire continues to evolve profoundly as the B-1a cells divide over the life of the animal. On the other hand, the gene rearrangements that make the heavy chains in the B-2 cells continue throughout the life of the animal to produce the wider repertoire of antibodies found in these cells. In addition, the processes that continue to change the antibody reperotire in the B-1a cells during adulthood do not occur in the B-2 populations.

Importantly, the these reperotire-changing processes in B-1a cells also occur in mice that have been raised in germ-free conditions, which demonstrates that – unlike other B cells – the repertoire of heavy chains in B-1a cells is not influenced by antigens from microbes. Instead, it is mainly driven by antigens that are expressed by normal cells in the body. These findings open the way to future work aimed at understanding how B-1a cells help to protect us against infection, and their role in autoimmune diseases, where immune cells attack the body's own healthy cells.

the center of antigen binding site and plays a crucial role in defining antibody specificity and affinity (*Xu and Davis, 2000*).

After encountering antigen, "'naïve"' B cells are activated and can further diversify their primary antibody repertoire by activation-induced cytidine deaminase (AID)–mediated somatic hypermutation (SHM), which introduces single or multiple mutations into the IgV regions (*Muramatsu et al., 2000*; *Wagner and Neuberger, 1996*). SHM commonly occurs in germinal centers (GC) (*Victora and Nussenzweig, 2012*), where memory B cells expressing high affinity antibodies are selected (*Rajewsky, 1996*; *Gitlin et al., 2014*). Since the antigen-driven SHM-mediated secondary Ig diversification is viewed as a crucial adaptation to the environmental needs, the IgH repertoire (s) expressed by FOB, MZB and B-1a cells from non-immunized animals are thought to be free of SHM. Our studies here, however, introduce a previously unrecognized SHM mechanism that increasingly diversifies the B-1a pre-immune IgH repertoire as animals age. Importantly, the SHM operates equally in the presence or absence of microbiota influence.

The B-1a antibody repertoire is commonly thought to be 'restricted' with expressing germline genes, largely because the hybridomas generated from fetal and neonatal B cells, which are mainly

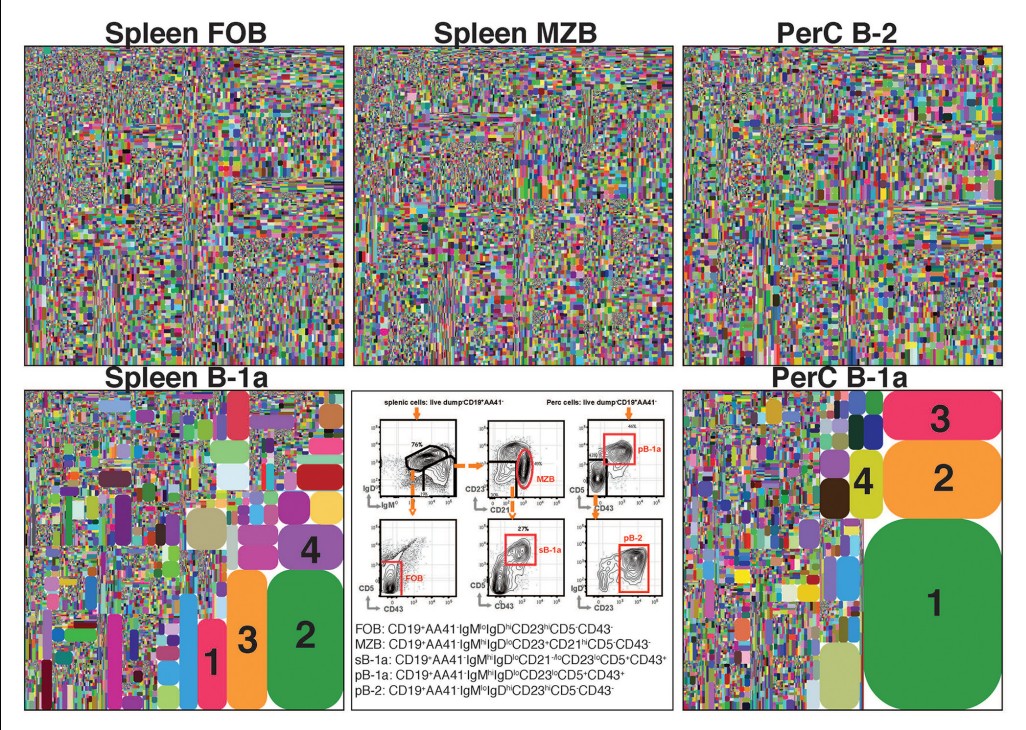

**Figure 1.** The B-1a IgH CDR3 sequences are much less diverse and recur more frequently than the CDR3 sequences expressed by FOB and MZB B subsets. IgH CDR3 tree-map plots illustrating the IgH CDR3 nucleotide sequences expressed by indicated B cell subsets sorted from one 2-month old C57Bl/6 mouse. Each rectangle in a given tree-map represents a unique CDR3 nucleotide sequence and the size of each rectangle denotes the relative frequency of an individual sequence. The colors for the individual CDR3 sequences in each tree-map plot are chosen randomly thus do not match between plots. The numbers shown in the CDR3 tree-map plots highlight the highly reoccurring CDR3 sequences including PtC-binding CDR3 sequences. 1, ARFYYYGSSYAMDY, V1-55D1-1J4; 2, MRYGNYWYFDV, V11-2D2-8J1; 3, MRYSNYWYFDV, V11-2D2-6J1; 4, MRYGSSYWYFDV, V11-2D1-1J1. *Lower middle panel*: FACS plots showing the gating strategy used to sort the phenotypically defined each B cell subset from spleen (s) or peritoneal cavity (p). Note: peritoneal B-1a cells are well known to express CD11b, a marker expressed on many myeloid cells including macrophage and neutrophils. The level of CD11b expressed on peritoneal B-1a cells, however, is roughly 100 fold lower than the level of CD11b expressed on the myeloid cells. This drastic difference is sufficient to separate the CD11b$^+$ B-1a cells from the myeloid cells if monoclonal anti-CD11b reagent is included in the dump channel (*Figure 1—figure supplement 3*).

The following figure supplements are available for figure 1:

**Figure supplement 1.** FACS plots showing CD43$^+$ CD5$^+$ IgM$^+$ B-1a cells in E19 fetal liver.

**Figure supplement 2.** Recurrent V$_H$11-encoded PtC-binding V(D)J sequences.

**Figure supplement 3.** CD11b expression on peritoneal B-1a (CD5$^+$) and B-1b (CD5$^-$) is roughly 100-fold lower than the CD11b expression on myeloid cells.

---

B-1a, have few N-insertions (*Carlsson and Holmberg, 1990*) and preferentially express the proximal 7183, Q52 V$_H$ family genes (*Perlmutter et al., 1985*). The N diversity deficit is ascribed to the absence of expression of terminal deoxynucleotidyl transferase (*Tdt*), which adds the N nucleotides to the CDR3 junction (*Gilfillan et al., 1993*), during fetal life (*Feeney, 1990*). These early studies left the impression that the proximal V$_H$ gene usage predominates and that there is little N-addition in the B-1a IgH repertoire.

Later studies by the Rajewsky group, however, showed that although neonatal (4 day) splenic B-1a cells contain very few N-insertions, N addition is readily detected in substantial numbers of peritoneal B-1a cells from adult animals (*Gu et al., 1990*), indicating that B-1a cells are continuously

**Table 1.** Summary of the sequences for 60 separately sorted B cell populations analyzed in this study.

| Sample | Id | Subset | Strain | Age | Condition | Mice | RNT* | RNU* | RPU* | CNT* | CNU* | CPU* |
|---|---|---|---|---|---|---|---|---|---|---|---|---|
| 1 | 7631 | FOB | WT | 2M | SPF | single | 1006030 | 151210 | 65871 | 903400 | 21240 | 20470 |
| 2 | 13966 | FOB | WT | 3.5M | SPF | single | 150812 | 31003 | 24801 | 130652 | 14911 | 14678 |
| 3 | 8706 | FOB | WT | 4M | SPF | single | 180365 | 53577 | 27817 | 159710 | 16901 | 16568 |
| 4 | 8702 | FOB | WT | 5M | SPF | single | 156681 | 54195 | 27728 | 136101 | 16951 | 16649 |
| 5 | 13967 | FOB | AID KO | 5M | SPF | single | 35967 | 14623 | 13203 | 27726 | 7187 | 7133 |
| 6 | 11161 | MZB | WT | 1M | SPF | single | 33548 | 19628 | 12744 | 25674 | 6584 | 6471 |
| 7 | 10658 | MZB | WT | 2M | SPF | single | 71458 | 26978 | 18278 | 61258 | 11512 | 11170 |
| 8 | 7630 | MZB | WT | 2M | SPF | single | 1032381 | 139832 | 62520 | 932353 | 20780 | 19792 |
| 9 | 8701 | MZB | WT | 4M | SPF | single | 214238 | 55075 | 26458 | 191065 | 15461 | 15021 |
| 10 | 8700 | MZB | WT | 5M | SPF | single | 118863 | 42310 | 22794 | 102894 | 14517 | 14180 |
| 11 | 13338 | MZB | WT | 4M | GF | single | 162754 | 39930 | 23611 | 141646 | 12939 | 12605 |
| 12 | 13343 | MZB | WT | 4M | GF | single | 595780 | 85497 | 45820 | 536072 | 19266 | 18480 |
| 13 | 11163 | pB-1a | WT | 1M | SPF | pool of 3 mice | 45882 | 11290 | 5596 | 41368 | 3237 | 3007 |
| 14 | 10660 | pB-1a | WT | 2M | SPF | single | 222324 | 17311 | 8630 | 207749 | 3891 | 3649 |
| 15 | 13018 | pB-1a | WT | 2M | SPF | single | 808879 | 36031 | 14817 | 753868 | 4769 | 4374 |
| 16 | 7628 | pB-1a | WT | 2M | SPF | single | 1784677 | 59458 | 22105 | 1706235 | 6601 | 5848 |
| 17 | 11160 | pB-1a | WT | 2W | SPF | pool of 8 mice | 65317 | 14700 | 7025 | 58034 | 4240 | 3704 |
| 18 | 10655 | pB-1a | WT | 3W | SPF | pool of 5 mice | 62875 | 12162 | 6622 | 57558 | 4180 | 3694 |
| 19 | 8705 | pB-1a | WT | 4M | SPF | single | 310077 | 28441 | 11886 | 287695 | 5063 | 4707 |
| 20 | 9870 | pB-1a | WT | 4M | SPF | single | 229100 | 26299 | 10469 | 211514 | 4745 | 4480 |
| 21 | 11165 | pB-1a | WT | 5M | SPF | single | 105410 | 19528 | 8926 | 95994 | 4435 | 4162 |
| 22 | 8707 | pB-1a | WT | 5M | SPF | single | 320252 | 29786 | 12423 | 296946 | 4722 | 4384 |
| 23 | 9861 | pB-1a | WT | 6M | SPF | single | 26613 | 5683 | 3235 | 23542 | 1521 | 1461 |
| 24 | 8704 | pB-1a | AID KO | 4M | SPF | single | 264340 | 33745 | 14519 | 245941 | 6648 | 6294 |
| 25 | 10657 | pB-2 | WT | 2M | SPF | single | 53953 | 23059 | 16883 | 44986 | 10084 | 9923 |
| 26 | 7629 | pB-2 | WT | 2M | SPF | single | 1315663 | 123472 | 47337 | 1238225 | 16925 | 16065 |
| 27 | 13969 | pB-2 | WT | 3.5M | SPF | single | 186817 | 24304 | 17689 | 170768 | 9089 | 8925 |
| 28 | 9862 | pB-2 | WT | 4M | SPF | single | 22591 | 13377 | 8737 | 17343 | 4382 | 4357 |
| 29 | 13973 | pB-2 | AID KO | 5M | SPF | single | 617893 | 62319 | 41165 | 566826 | 17536 | 16965 |
| 30 | 13000 | sB-1a | WT | 2d | SPF | pool of 8 mice | 29439 | 9542 | 4925 | 25369 | 3148 | 2758 |
| 31 | 10651 | sB-1a | WT | 5d | SPF | single | 123360 | 22472 | 10838 | 113161 | 7453 | 5976 |
| 32 | 10659 | sB-1a | WT | 5d | SPF | single | 210055 | 28140 | 12411 | 192662 | 7307 | 5812 |
| 33 | 9866 | sB-1a | WT | 5d | SPF | single | 52986 | 15600 | 6864 | 46580 | 4595 | 3837 |
| 34 | 10652 | sB-1a | WT | 6d | SPF | single | 172875 | 26437 | 12545 | 159304 | 7683 | 6365 |
| 35 | 9865 | sB-1a | WT | 7d | SPF | single | 71309 | 18446 | 8775 | 64241 | 5482 | 4941 |
| 36 | 9868 | sB-1a | WT | 7d | SPF | single | 201813 | 35069 | 14473 | 186227 | 7847 | 6843 |

*Table 1 continued on next page*

*Table 1 continued*

| # | Id | | | | | | RNT* | RNU* | RPU* | CNT* | CNU* | CPU* |
|---|---|---|---|---|---|---|---|---|---|---|---|---|
| 37 | 10656 | sB-1a | WT | 2M | SPF | single | 369732 | 39603 | 19759 | 342914 | 9489 | 9048 |
| 38 | 13004 | sB-1a | WT | 2M | SPF | single | 185948 | 27952 | 13875 | 168522 | 7313 | 7022 |
| 39 | 7632 | sB-1a | WT | 2M | SPF | single | 1825218 | 102797 | 43190 | 1719246 | 12428 | 11144 |
| 40 | 11168 | sB-1a | WT | 2W | SPF | single | 536603 | 70201 | 28829 | 496671 | 11948 | 10913 |
| 41 | 13005 | sB-1a | WT | 2W | SPF | single | 98017 | 28331 | 15001 | 85489 | 8820 | 8207 |
| 42 | 10654 | sB-1a | WT | 3W | SPF | single | 146560 | 33814 | 19697 | 131091 | 11995 | 11451 |
| 43 | 13970 | sB-1a | WT | 3.5M | SPF | single | 170925 | 13809 | 9289 | 160480 | 4513 | 4273 |
| 44 | 13335 | sB-1a | WT | 4M | SPF | single | 22175 | 4822 | 3449 | 18683 | 1131 | 1090 |
| 45 | 13342 | sB-1a | WT | 4M | SPF | single | 283072 | 23668 | 12947 | 262744 | 5357 | 5032 |
| 46 | 8699 | sB-1a | WT | 4M | SPF | single | 142838 | 19151 | 9938 | 130915 | 4370 | 4086 |
| 47 | 9863 | sB-1a | WT | 4M | SPF | single | 73676 | 16599 | 8713 | 65571 | 4233 | 4092 |
| 48 | 11167 | sB-1a | WT | 5M | SPF | single | 501367 | 38912 | 17336 | 463863 | 7573 | 7163 |
| 49 | 8708 | sB-1a | WT | 5M | SPF | single | 577114 | 52723 | 22272 | 531508 | 9146 | 8441 |
| 50 | 9867 | sB-1a | WT | 6M | SPF | single | 113492 | 20612 | 10625 | 101791 | 4563 | 4343 |
| 51 | 13965 | sB-1a | AID KO | 4M | SPF | single | 177782 | 16419 | 12281 | 164189 | 6539 | 6293 |
| 52 | 13971 | sB-1a | AID KO | 4M | SPF | single | 517141 | 34159 | 22031 | 482543 | 8966 | 8395 |
| 53 | 13968 | sB-1a | AID KO | 5M | SPF | single | 427671 | 30839 | 20510 | 396974 | 9162 | 8545 |
| 54 | 13972 | sB-1a | AID KO | 5M | SPF | single | 706116 | 36217 | 23255 | 660874 | 9294 | 8744 |
| 55 | 13001 | sB-1a | WT | 4M | GF | single | 43507 | 8734 | 4855 | 38947 | 2318 | 2249 |
| 56 | 13002 | sB-1a | WT | 4M | GF | single | 47203 | 8683 | 4820 | 42279 | 2053 | 1965 |
| 57 | 13003 | sB-1a | WT | 4M | GF | single | 213347 | 22246 | 11068 | 197769 | 4705 | 4449 |
| 58 | 13017 | sB-1a | WT | 4M | GF | single | 532250 | 40497 | 17375 | 501908 | 7019 | 6398 |
| 59 | 13337 | sB-1a | WT | 4M | GF | single | 28559 | 6322 | 4417 | 24047 | 1544 | 1486 |
| 60 | 13341 | sB-1a | WT | 4M | GF | single | 388208 | 28942 | 14837 | 360727 | 5674 | 5144 |

Id is a unique identifier for the sequence run

RNT*, total raw nucleotide sequences

RNU*, unique raw nucleotide sequences

RPU*, unique raw peptide sequences

CNT*, total clean nucleotide sequences

CNU*, unique clean nucleotide sequences

CPU*, unique clean peptide sequences

| Sequence statistics | RNT* | RNU* | RPU* | CNT* | CNU* | CPU* |
|---|---|---|---|---|---|---|
| Total | 1.9E + 07 | 2.1E + 06 | 1.1E + 06 | 1.8E + 07 | 4.9E + 05 | 4.7E + 05 |
| Mean | 319865 | 35610 | 17848 | 295174 | 8233 | 7762 |
| % CV | 122 | 86 | 74 | 125 | 61 | 63 |

generated after *Tdt* is expressed. Holmberg lab similarly found the low N-region diversity in the adult peritoneal B-1a repertoire (*Tornberg and Holmberg, 1995*). Our early studies confirm and extend these findings by showing that roughly two thirds of the IgH sequences from individually sorted peritoneal B-1a cells have N additions (*Kantor et al. 1997*). Furthermore, recent studies have shown that B-1a progenitors from both fetal liver and adult BM sources generate peritoneal B-1a cells with substantial N-addition (*Holodick et al., 2014*). Collectively, these findings demonstrate that the peritoneal B-1a IgH repertoire diversity is greater than previously thought.

However, these studies mainly characterized the repertories of B cells in the peritoneal cavity (PerC) and leave the questions open as to whether and how the repertoire changes throughout ontogeny in B cells at various sites of development and function. Studies here address these issues. We show that the B-1a IgH repertoire differs drastically from the repertories expressed by splenic FOB, MZB and peritoneal B-2 cells. In addition, we track the development of B-1a cells from their early appearance in neonatal spleen to their long-term residence in adult peritoneum and spleen, and elucidate the previous unrecognized somatic mechanisms that select and diversify the B-1a IgH repertoire over time. Most importantly, the potent mechanisms that uniquely act in B-1a (not in FOB and MZB cells) operate comparably in germ-free (GF) and conventional mice reared under specific pathogen free (SPF) condition, indicating that these repertoire-defining mechanisms are not driven by microbiota-derived antigens.

The dearth of these advanced understandings in the previous studies is largely due to technical difficulties that limited both their scope and depth. Studies analyzing Ig sequences from immortalized cell lines (e.g., hybridomas) or LPS-stimulated B cells had obvious sampling biases. In addition, earlier studies mainly focused on particular $V_H$ families (e.g., J558, 7183), even though the mouse IgH locus contains over 100 functional $V_H$ genes (*Kirkham and Schroeder, 1994*). The introduction of single cell analyses enabled higher precision and lower bias than the bulk measurements. However, they were constrained profoundly by sequencing costs and technical challenges. Indeed, our previous single cell analysis reported only 184 IgH sequences derived from 85% recovered sorted single cells representative of three types of peritoneal B subsets (*Kantor et al., 1997*). Thus, while the data yielded key insights, hundreds or thousands of single cells would need to be analyzed to obtain a more comprehensive view for a single B subset repertoire. Finally, difficulties in defining and cleanly sorting rare B subsets (e.g., splenic B-1a) further compromise the attempt to develop a thorough view of repertoire(s) expressed by various B cell subsets at the different anatomic location and ontogenic stage.

To overcome these obstacles, we have coupled high-dimensional (Hi-D) FACS sorting with unique IgH multiplex PCR technologies, which allow inclusive amplification of IgH transcripts for each sorted B subset and ultimate sequencing of these sequences. Using barcoded sample multiplexing, we have performed a large-scale quantitative and comparative study of the 'pre-immune' IgH repertoires expressed by various functionally and developmentally distinct mature B subsets (splenic FOB, MZB and B-1a; peritoneal B-2 and B-1a) from non-immune C57BL/6J mice. In addition, since microbiota are often thought to influence the Ig repertoire, we have compared the B-1a IgH repertoires in GF or conventional mice.

## Results

### The B-1a pre-immune IgH repertoire is far more restricted and repetitive than the repertoire expressed by FOB and MZB subsets

We sorted splenic and peritoneal B-1a (dump$^-$ CD19$^+$ CD93$^-$IgM$^{hi}$ IgD$^{lo/-}$ CD21$^{-/lo}$ CD23$^-$ CD43$^+$ CD5$^+$); splenic FOB and peritoneal B-2 (dump$^-$ CD19$^+$ CD93$^-$ IgM$^{lo}$ IgD$^{hi}$ CD23$^+$ CD43$^-$ CD5$^-$); and splenic MZB (dump$^-$ CD19$^+$ CD93$^-$ IgM$^{hi}$ IgD$^{lo/-}$ CD21$^{hi}$ CD23$^{lo/-}$ CD43$^-$ CD5$^-$) from non-immune C57BL/6 mice (*Figure 1*). We generated and amplified IgH cDNA libraries from each subset. We then pooled the libraries, which are distinguishable by barcode, and sequenced them (Illumina MiSeq). In all, we sequenced 60 separately prepared libraries, each derived from 1-2 x10$^4$ B cells of a given subset sorted from mice at the same or different ages (from 2 days to 6 months, > 30 mice) (*Table 1*). Overall 18 million total clean nucleotide sequences (CNT) and about half million unique clean nucleotide sequences (CNU) were analyzed in the study (*Table 1*).

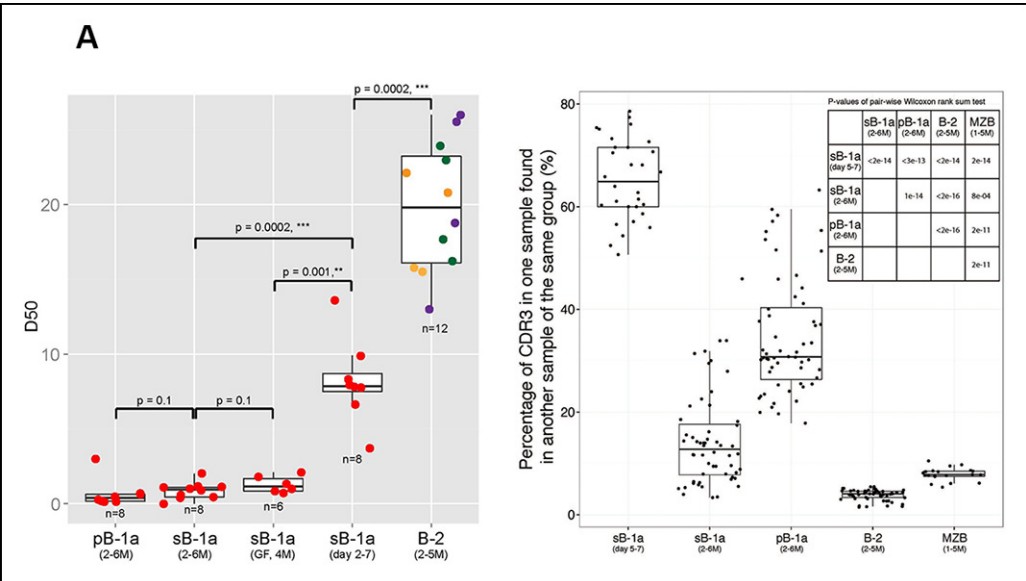

**Figure 2.** The B-1a pre-immune IgH repertoire is far more restricted than the pre-immune IgH repertoires expressed by splenic FOB, MZB and peritoneal B-2 cells. (**A**) D50 metric analysis quantifying the IgH CDR3 diversity for B cell subsets from mice at the indicated age. Low D50 values are associated with less diversity. Each dot represents the data for a B cell sample from an individual mouse except for the 2 day splenic B-1a data, which are derived from sorted cells pooled from 8 mice. B-1a samples are labeled with red; B-2 samples include FOB (green, n = 4), pB-2 (purple, n = 4) and MZB (yellow, n = 4). The data for germ-free (GF) animals is discussed at the end of the Result section. (**B**) CDR3 peptide pair-wise sharing analysis of IgH repertoire similarity among multiple samples for each B cell group (n = 5-9). Each dot represents the percentage of common CDR3 peptides in one sample that are also found in another sample within a given group. For example, to compute the similarity between sample A and B, the percentage of CDR3 peptides in sample A that are also found in sample B ($p_{A \longrightarrow B}$), together with the percentage of CDR3s in sample B that are also in sample A ($p_{B \longrightarrow A}$) are used as an indicator. For comparison of 6 splenic B-1a samples in 5-7 day group, there are 30 comparisons. *Right upper:* p values showing the statistical significance between two groups. Box plots represent the 10th, 25th, 50th, 75th and 90th percentiles here and in other figures.

We also attempted to analyze the B-1a repertoire in fetal liver but found that there were too few B-1a cells to reliably sequence with our method. In essence, FACS analysis of embryonic day 19 (E19) fetal liver cells shows that IgM$^+$ B cells represent only 0.6% of CD19$^+$ total B cells and that only around 20% of these IgM$^+$ B cells express the B-1a CD43$^+$ CD5$^+$ phenotype (*Figure 1—figure supplement 1*). The frequencies of IgM$^+$ B cell in E18 fetal liver are even lower (0.2% of CD19$^+$ B cells). These numbers are too low for us to recover enough material for sequencing from a feasible number of embryos.

The IgH CDR3 tree maps for each B cell subset show that splenic FOB and peritoneal B-2 cells express highly diversified IgH CDR3 nucleotide sequences, as do MZB cells (*Figure 1*). In contrast, CDR3 nucleotide sequences expressed by B-1a cells from either spleen or PerC are far less diverse and recur much more frequently (*Figure 1*). The recurrent CDR3 sequences include the well-studied V$_H$11-encoded sequences specific for phosphatidylcholine (PtC) (*Figure 1—figure supplement 2*) and known to occur frequently in B-1a cells (*Mercolino et al., 1988*; *Hardy et al., 1989*; *Seidl et al., 1997*).

D50 metric analysis quantifying the IgH CDR3 nucleotide sequence diversity shows that the IgH CDR3 nucleotide sequences expressed by the FOB and MZB subsets are significantly more diverse than those expressed by splenic and peritoneal B-1a cells (p = 0.0002, Mann-Whitney-Wilcoxon Test) (*Figure 2A*). Consistent with this finding, IgH CDR3 peptide pairwise sharing analysis, which measures the similarity of IgH CDR3 peptide expression for each B cell subset sorted from different mice, shows that the same CDR3 peptide sequences frequently appear in both splenic and peritoneal B-1a cells from different mice whereas the common CDR3 peptides are rare in FOB and MZB

**Table 2.** Top 10 highly recurring CDR3 sequences (peptide and V(D)J recombination) detected in each of the listed splenic B-1a samples.

| sB-1a samples | | Top 10 IgH CDR3 sequences | | |
|---|---|---|---|---|
| Id | Age | | Peptide | V(D)J |
| 11168 | 2 weeks | 1 | ANDY | V1-53 J2 |
| | | 2 | AKHGYDAMDY | V2-9 D2-9 J4 |
| | | 3 | ARRYYGSSYWYFDV | V1-55 D1-1 J1 |
| | | 4 | ANWDY | V1-53 D4-1 J2 |
| | | 5 | MRYSNYWYFDV | V11-2 D2-6 J1 |
| | | 6 | ARDAYYWYFDV | V7-1 J1 |
| | | 7 | ATDYYAMDY | V1-26 J4 |
| | | 8 | ARFYYYGSSYAMDY | V1-55 D1-1 J4 |
| | | 9 | AIYYLDY | V1-53 D2-8 J2 |
| | | 10 | ARHYGSSYWYFDV | V2-6-2 D1-1 J1 |
| 10654 | 3 weeks | 1 | ARRYYGSSYWYFDV | V1-55 D1-1 J1 |
| | | 2 | ARSYSNYVMDY | V1-76 D2-6 J4 |
| | | 3 | ARYYGSNYFDY | V7-3 D1-1 J2 |
| | | 4 | ARGASYYSNWFAY | V1-55 D2-6 J3 |
| | | 5 | ALTGTAY | V1-53 D4-1 J3 |
| | | 6 | ARAGAGWYFDV | V5-9 D4-1 J1 |
| | | 7 | TYSNY | V6-6 D2-6 J2 |
| | | 8 | ARTGTYYFDY | V1-53 D4-1 J2 |
| | | 9 | AMVDY | V1-64 D2-9 J2 |
| | | 10 | ARWGTTVVGY | V1-7 D1-1 J2 |
| 7632 | 2 months | 1 | MRYGNYWYFDV | V11-2 D2-8 J1 |
| | | 2 | MRYSNYWYFDV | V11-2 D2-6 J1 |
| | | 3 | MRYGSSYWYFDV | V11-2 D1-1 J1 |
| | | 4 | ATFSY | V1-55 J2 |
| | | 5 | ARFYYYGSSYAMDY | V1-55 D1-1 J4 |
| | | 6 | ARIPNWVWYFDV | V1-55 D4-1 J1 |
| | | 7 | ARWDTTVVAPYYFDY | V1-7 D1-1 J2 |
| | | 8 | ARDYYGSSWYFDV | V1-26 D1-1 J1 |
| | | 9 | TYYDYDLYAMDY | V14-4 D2-4 J4 |
| | | 10 | ARFITTVVATRYWYFDV | V1-9 D1-1 J1 |
| 8699 | 4 months | 1 | ARSADYGGYFDV | V1-64 D2-4 J1 |
| | | 2 | ARGAY | V1-80 J2 |
| | | 3 | ARSYYDYPWFAY | V1-76 D2-4 J3 |
| | | 4 | ARRWLLNAMDY | V1-9 D2-9 J4 |
| | | 5 | ARPYYYGSSPWFAY | V1-69 D1-1 J3 |
| | | 6 | ARNDYPYWYFDV | V1-4 D2-4 J1 |
| | | 7 | ARSGDY | V1-64 J2 |
| | | 8 | ARVIGDY | V1-53 D2-14 J4 |
| | | 9 | ARANY | V1-55 J3 |
| | | 10 | AVNWDYAMDY | V1-84 D4-1 J4 |

*Table 2 continued on next page*

*Table 2 continued*

| sB-1a samples | | Top 10 IgH CDR3 sequences | | |
|---|---|---|---|---|
| Id | Age | | Peptide | V(D)J |
| 8708 | 5 months | 1 | ASLTY | V1-55 J2 |
| | | 2 | TCNYH | V14-4 D2-8 J4 |
| | | 3 | LIGRNY | V1-55 D2-14 J2 |
| | | 4 | MRYSNYWYFDV | V11-2 D2-6 J1 |
| | | 5 | AKQPYYGSSYWYFDV | V2-3 D1-1 J1 |
| | | 6 | AGSSYAYYFDY | V1-66 D1-1 J2 |
| | | 7 | ARRGIDLLWYHYYAMDY | V1-26 D2-8 J4 |
| | | 8 | ARKSSGSRAMDY | V7-3 D3-2 J4 |
| | | 9 | ASYAMDY | V7-3 J4 |
| | | 10 | ARLYYGNSYWYFDV | V1-55 D2-8 J1 |
| 9867 | 6 months | 1 | ARKYYPSWYFDV | V1-55 D1-1 J1 |
| | | 2 | AREGGKFY | V1-7 J2 |
| | | 3 | AKSSGYAMDY | V1-55 D3-2 J4 |
| | | 4 | ARWVITTVARYFDV | V1-85 D1-1 J1 |
| | | 5 | ARGFY | V1-80 J2 |
| | | 6 | AKEGGYYVRAMDY | V1-55 D1-2 J4 |
| | | 7 | ARSMDY | V1-80 J4 |
| | | 8 | ASAMDY | V1-64 J4 |
| | | 9 | TKGGYHDYDDGAWFVY | V1-53 D2-4 J3 |
| | | 10 | ARKFYPSWYFDV | V1-55 J3 |

Table lists the top 10 highly recurring CDR3 sequences (peptide and V(D)J recombination) shown in the individual CDR3 tree-map plot of the splenic B-1a samples from 2 week to 6 month old mice (**Figure 5A**). For each splenic B-1a sample, the Id number and mouse age are shown in column 1 and column 2 respectively.

subsets (*Figure 2B*). Taken together, these data demonstrate that the B-1a pre-immune IgH repertoire is far more restricted and repetitive than IgH repertoires expressed by FOB and MZB subsets.

## $V_H$ gene usage differs among the B-1a, FOB and MZB pre-immune IgH repertoires

We quantified the frequency of IgH sequences expressing individual $V_H$ gene for each sorted B cell sample and then compared the $V_H$ gene usage between two B cell subsets. B-1a cells are well-known to undergo self-replenishing in adult (*Kantor et al., 1995*). To minimize the impact of clonal expansion on the $V_H$ gene usage profile, we collected normalized data, in which we scored each distinct IgH CDR3 nucleotide sequence expressing a given $V_H$ gene as one, no matter how many times this sequence was detected.

Our approach enables detection of Ig transcripts expressing about 100 different $V_H$ genes that belong to 14 $V_H$ families (*Figure 3*). B-1a cells express all of these detected $V_H$ genes (*Figure 3A*), contrasting with earlier impressions, based largely on hybridomas sequences from fetal and neonatal mice (*Malynn et al., 1990*), that $V_H$ usage in the B-1a repertoire is very restricted. However, despite the broad $V_H$ usage, certain $V_H$ genes, notably V10-1 (DNA4), V6-6 (J606), V11-2 ($V_H$11) and V2-6-8 (Q52), are expressed at a significantly higher frequency in splenic B-1a than MZB cells ($p<0.05$, Welch's t-test, *Figure 3B*).

Similar to MZB cells, splenic FOB and peritoneal B-2 cells show lower frequency in expressing these B-1a favored $V_H$ genes, i.e., V6-6 (J606), V11-2 ($V_H$11) and V2-6-8 (Q52) (*Figure 3—figure supplement 1B–C*). Conversely, these B subsets tend to preferentially use the largest $V_H$ family, V1 (J558), located distal to $D_H$ and $J_H$ gene segments (*Yancopoulos and Alt, 1986*). MZB cells, in

particular, have a higher tendency to express certain V1 (J558) family genes including V1-82, V1-72, V1-71, V1-42, V1-18 and V1-5 (*Figure 3B*).

The $V_H$ usage in the peritoneal B-1a cells is further biased toward V6-6 (J606), V9-3 (Vgam3.8), V2-9 (Q52) and V2-6-8 (Q52) genes, which are already favored in the splenic B-1a cells (*Figure 3—figure supplement 1A*). This finding indicates that the splenic and peritoneal B-1a populations are not in equilibrium and the latter is further enriched for cells expressing certain $V_H$ genes.

## The B-1a IgH repertoire integrates rearrangements from de novo B-1a development that occur mainly during the first few weeks of life

Unlike FOB and MZB subsets, *de novo* B-1a development initiates prior to birth and decreases to a minimum in adult animals (*Lalor et al., 1989*; *Barber et al., 2011*). B-1a cells persist thereafter as a self-replenishing population (*Kantor et al., 1995*). To minimize the impact of self-replenishment on the N-addition distribution profile, and hence to weight the repertoire for de novo generated IgH sequences for B-1a cells, we collected normalized data that counts each distinct IgH sequence containing indicated N nucleotide insertions as a single sequence, regardless how many times this sequence was detected.

Consistent with *Tdt* expression, which is absent during the fetal life and initiates shortly after birth (*Feeney, 1990*; *Bogue et al., 1992*), N nucleotide insertion analysis of the splenic B-1a IgH repertoires demonstrate that roughly 60% of IgH sequences expressed by splenic B-1a cells from 2-–6 day mice do not contain N insertions at IgH CDR3 junction (D-J and V-DJ); about 30% contain 1–2 insertions; and, <15% contain 3–4 N-nucleotide insertions (*Figure 4A,B*). After 6 days, however, the frequency of sequences containing >3 N-additions progressively increases until the animals are weaned (roughly 3 weeks) (*Figure 4A,B*). After weaning, the N-addition pattern stabilizes, i.e., about 50% IgH sequences contain 3–7 N nucleotide insertions and about 30% have more than 8 N nucleotide insertions at IgH CDR3 junctions, and remains stable at this level for at least 5 months (*Figure 4A,B*).

In essence, splenic B-1a cells from 2-6 day mice largely originate from fetal and early neonatal wave(s) of B-1a development when *Tdt* is poorly expressed. As newborns progress to maturity, B-1a cells, which are originated in the earlier wave(s), are 'diluted' by B-1a cells that emerge during later development. The high frequency of N nucleotide additions in the adult splenic B-1a IgH repertoire indicates that a higher proportion of B-1a cells are actually generated postnatally after *Tdt* is expressed.

Cohering with the increased N diversity in the adulthood, CDR3 peptide pairwise sharing analysis shows that the expression of common IgH CDR3 peptides is significantly more frequent in neonatal splenic B-1a cells than in adult splenic B-1a cells (p<2e-16, Mann-Whitney-Wilcoxon Test, *Figure 2B*). $V_H$ usage also shifts as animals mature. Splenic B-1a cells from neonatal mice (2-–7 days) preferentially express the V3 (36–60), V5 (7183) and V2 (Q52) families that are largely located proximal to D and J gene segments (*Figure 3—figure supplement 1D*), consistent with previous findings that hybridomas derived from fetal/neonatal B cells are bias in expressing proximal V5 (7183) and V2 (Q52) family genes (*Perlmutter et al., 1985*). In contrast, the splenic B-1a cells from adult animal (2–6 months) show higher frequencies in expressing distal V1 (J558) family genes including V1-75, V1-64, V1-55 and V1-53 (*Figure 3—figure supplement 1D*).

Collectively, we conclude that the B-1a IgH repertoire integrates rearrangements from sequential waves of de novo B-1a development that mainly occur during the first few weeks of life. The IgH repertoires defined during these waves are distinguishable both by N-additions at CDR3 junctions and by $V_H$ gene usage.

## Recurring V(D)J sequences increase with age in the pre-immune B-1a IgH repertoire

Certain V(D)J nucleotide sequences become progressively more dominant with age in the B-1a repertoire. Thus, only a lower proportion of V(D)J sequences are detected at relative higher frequency in the splenic B-1a IgH repertoire before 3 weeks, after which, both the number of recurrent sequences and the frequency at which each is represented increase progressively until the animals reach 4–6 month of age (*Figure 5A*, *Table 2*). Consequently, the distribution of the splenic B-1a IgH CDR3

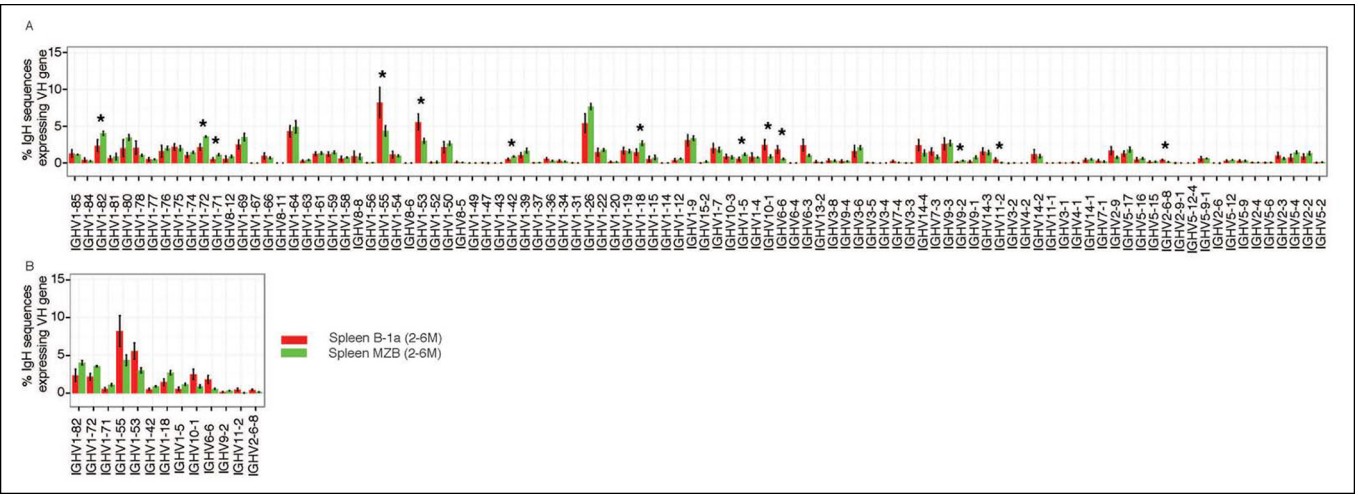

**Figure 3.** Comparison of $V_H$ gene usage by splenic B-1a vs MZB B cells. (**A**) $V_H$ gene usage profile shown as the percentage of IgH sequences expressing the listed individual $V_H$ genes for individual B cell samples. The profiles are shown for adult splenic B-1a samples (n = 9, red) and for MZB samples (n = 5, green). $V_H$ genes (from left to right) are ordered in 5'- to 3'-direction bases on chromosome location; the IMGT $V_H$ gene nomenclature is used (*Lefranc, 2003*). (**B**) $V_H$ genes showing the statistically significant differences (Welch's t-test p<0.05) between two groups are listed and also highlighted with asterisks in the plot. To minimize the impact of the clonal expansion on the $V_H$ gene usage profile, data are presented as the normalized distribution that counts each distinct CDR3 nucleotide sequence expressing a given $V_H$ gene as one, no matter how many times the sequence was detected. Note: $V_H$12-3 encoded IgH sequences are not detected in this study due to the technical limitations that exclude the $V_H$12-3 primer from the set of primers designed about three years ago and used for studies presented here. We have since corrected this problem so that $V_H$12-3 primer is now part of our new set of primers. Comparison of sequence data obtained with old vs. the new set of primers shows that, aside from now detecting $V_H$12-3 sequences with the new set of primers, the sequences obtained with both primer sets are highly similar (*Figure 3—figure supplement 2*).

The following figure supplements are available for figure 3:

**Figure supplement 1.** $V_H$ gene usage profile pair-wise comparison of B cell groups.

**Figure supplement 2.** Almost identical top 10 highly recurring CDR3 sequences are detected for splenic B-1a IgH libraries obtained either with the old or new primer set.

nucleotide sequences diversity is much less random in adults (2–6 months) than in neonates (2–7 days) (*Figure 2A*).

The recurrent V(D)J sequences include $V_H$11-encoded PtC-binding V(D)J sequences, which are initially present at very low frequencies (2–6 days) but increase aggressively as animals mature to middle age (6 months) (*Figure 5B*). Since de novo B-1a development is minimum at adulthood, the progressive increase in the representation of the recurrent V(D)J sequences as animals reach adulthood suggests that B-1a cells are self-replenishing.

## Certain V(D)J sequences are conserved by being positively selected into the shared adult B-1a pre-immune IgH repertoire

To determine to what extent the IgH CDR3 sequences (amino acid and nucleotide) expressed by each B cell subset are shared across different individuals, we carried out CDR3 sharing analysis. In the B-1a IgH repertoire, overall, we found 30 such highly shared IgH CDR3 peptides, each of which is expressed in over 80% of the splenic B-1a samples taken from more than 20 animals with nine different ages (from 2 days to 6 months) (*Table 3*). Each of the shared CDR3 peptides would be expected to be encoded by several convergent V(D)J recombinations, i.e., distinct V(D)J rearrangements encode the same CDR3 amino acid sequence (*Venturi et al., 2008*). Strikingly, we found that each of the shared CDR3 peptides is encoded by an identical V(D)J nucleotide sequence in over 70% of splenic B-1a samples from *adult* animals (2-6 months, 9 mice) (*Table 3*).

These V(D)J nucleotide sequences represent the IgH structures that are positively selected into the shared adult B-1a IgH repertoire among C57BL/6 mice. Although the specificities of the majority

**Table 3.** Certain V(D)J sequences are positively selected and conserved in adult B-1a pre-immune IgH repertoires.

| CDR3 peptide | | | Predominant V(D)J | | CDR3 junction diversity | | Representation in indicated repertoire | | | |
|---|---|---|---|---|---|---|---|---|---|---|
| splenic B-1a (2d-6M) | | | splenic B-1a (2-6M) | | addition | deletion | PerC B-1a (2W-6M) | splenic B-1a (4M germ free) | FOB (2-5M) | MZB (1-5M) |
| 1 | TRWDY | 17/20 | V6-6 J2 | 8/9 | TGG | J2(8) | 11/11 | 5/6 | 1/8 | 0/7 |
| 2 | MRYSNYWYFDV | 17/20 | V11-2 D2-6 J1 | 9/9 | 0 | 0 | 11/11 | 6/6 | 1/8 | 1/7 |
| 3 | MRYGNYWYFDV | 18/20 | V11-2 D2-8 J1 | 9/9 | 0 | 0 | 11/11 | 6/6 | 1/8 | 1/7 |
| 4 | MRYGSSYWYFDV | 17/20 | V11-2 D1-1 J1 | 9/9 | 0 | 0 | 11/11 | 6/6 | 1/8 | 1/7 |
| 5 | VRHYGSSYFDY | 15/20 | V10-1 D1-1 J2 | 5/9 | 0 | J2(1) | 11/11 | 3/6 | 0/8 | 0/7 |
| 6 | ARHYYGSSYYFDY | 19/20 | V5-6-1 D1-1 J2 | 9/9 | 0 | 0 | 11/11 | 6/6 | 2/8 | 0/7 |
| 7 | ARLDY | 20/20 | V1-53 J2 | 7/9 | CTg/a | J2(8) | 10/11 | 4/6 | 0/8 | 1/7 |
| 8 | ARDYYGSSYWYFDV | 19/20 | V7-1 D1-1 J1 | 6/9 | 0 | V7-1(3) | 9/11 | 5/6 | 1/8 | 1/7 |
| 9 | ARDYYGSSWYFDV | 19/20 | V1-26 D1-1 J1 | 7/9 | G | J1(3) | 2/11 | 4/6 | 0/8 | 1/7 |
| 10 | ANWDY | 19/20 | V14-3 D4-1 J2 | 6/9 | 0 | V14-3(2)J2(8) | 5/11 | 2/6 | 0/8 | 0/7 |
| 11 | ATGTWFAY | 18/20 | V1-19 D4-1 J3 | 5/9 | 0 | V1-19(2) | 6/11 | 2/6 | 0/8 | 1/7 |
| 12 | ARYYYGSSYAMDY | 19/20 | V7-3 D1-1 J4 | 8/9 | 0 | V7-3(1)J4(4) | 10/11 | 3/6 | 3/8 | 3/7 |
| 13 | ARYSNYYAMDY | 18/20 | V1-39 D2-6 J4 | 6/9 | 0 | J4(2) | 8/11 | 1/6 | 0/8 | 0/7 |
| 14 | ARDFDY | 19/20 | V1-64 J2 | 6/9 | G | J2(3) | 1/11 | 3/6 | 1/8 | 1/7 |
| 15 | ARYYSNYWYFDV | 17/20 | V1-9 D2-6 J1 | 6/9 | 0 | 0 | 4/11 | 1/6 | 0/8 | 0/7 |
| 16 | ARYDYDYAMDY | 17/20 | V1-39 D2-4 J4 | 6/9 | 0 | J4(3) | 7/11 | 1/6 | 0/8 | 0/7 |
| 17 | ARHYYGSSYWYFDV | 18/20 | V2-6-2 D1-1 J1 | 6/9 | 0 | 0 | 6/11 | 2/6 | 1/8 | 3/7 |
| 18 | ARFYYYGSSYAMDY | 19/20 | V1-55 D1-1 J4 | 6/9 | T | J4(4) | 8/11 | 3/6 | 1/8 | 1/7 |
| 19 | ARWDFDY | 19/20 | V1-7 J2 | 6/9 | TGGG | J2(3) | 1/11 | 3/6 | 1/8 | 1/7 |
| 20 | ARGAY | 19/20 | V1-80 J3 | 5/9 | GGG | J3(8) | 7/11 | 6/6 | 1/8 | 1/7 |
| 21 | ARRFAY | 18/20 | V1-26 J3 | 7/9 | C/A | J3(8) | 9/11 | 3/6 | 1/8 | 1/7 |
| 22 | ARRDY | 18/20 | V1-55 J2 | 5/9 | AGg/a | J2(8) | 6/11 | 3/6 | 1/8 | 1/7 |
| 23 | **ASYDGYYWYFDV** | 18/20 | V1-55 D2-9 J1 | 8/9 | CTATG | V1-55(1) | 9/11 | 5/6 | 0/8 | 0/7 |
| 24 | ASYAMDY | 16/20 | V7-3 J4 | 8/9 | 0 | V7-3(5)J4(4) | 9/11 | 6/6 | 0/8 | 1/7 |
| 25 | ARRYYFDY | 17/20 | V1-78 J2 | 7/9 | CGg/cT | 0 | 8/11 | 2/6 | 0/8 | 0/7 |
| 26 | ARNYYYFDY | 15/20 | V1-53 D1-2 J2 | 8/9 | t/a | 0 | 10/11 | 2/6 | 0/8 | 0/7 |
| 27 | ARYYGNYWYFDV | 15/20 | V3-8 D2-8 J1 | 5/9 | 0 | 0 | 5/11 | 2/6 | 0/8 | 0/7 |
| 28 | **ARRYYGSSYWYFDV** | 15/20 | V1-55 D1-1 J1 | 7/9 | CGG | 0 | 10/11 | 5/6 | 1/8 | 1/7 |
| 29 | ARRLDY | 13/20 | V1-22 J2 | 7/9 | CGAC | J2(6) | 8/11 | 2/6 | 0/8 | 1/7 |
| 30 | **ARFAY** | 18/20 | V1-80 J3 | 4/9 | 0 | J3(4) | 2/11 | 3/6 | 0/8 | 0/7 |

Column 1: CDR3 peptide sequences identified to be shared in >80% of splenic B-1a samples (20 samples from mice ranging from 2 day to 6 month old); Column 2: for each shared CDR3 peptide, a single V(D)Jrearrangement sequence is selected and conserved in over 70% of adult B-1a samples (9 samples, 2-6 month old); Columns 3 and 4: nucleotides added or deleted in CDR3 junctions; Columns 5-8: the representation of each selected V(D)J sequence within the indicate repertoires (age and number of samples are shown for each group). Rows 2-4 are PtC-binding CDR3 sequences; Row 8 is CDR3 sequence for T15 Id[+] anti-PC antibody. The data for germ-free animals is discussed at the end of the Result section.

of these selected V(D)J sequences remain to be defined, they include sequences that are specific for PtC and sequence for the T15 idiotype B-1a anti-PC antibodies (*Masmoudi et al., 1990*). Of note, most of these V(D)J sequences have nucleotide additions and/or deletions in the CDR3 junction (*Table 3*), indicating that the driving force for the selection may include, but is certainly not restricted to the germline rearrangement.

The majority of the V(D)J nucleotide sequences that are conserved in the splenic B-1a repertoire are also conserved in the peritoneal B-1a IgH repertoires (2W-6M, 11 samples) (*Table 3*). Such

V(D)J nucleotide sequences, however, are rarely detectable in FOB and MZB IgH repertoires (1-5M, 7-8 samples), either because these cells do not express these CDR3 peptides or because they use different V(D)J recombination sequences to encode them (*Table 3*). For example, although MZB cells express antibodies encoding the same CDR3 peptide as B-1a T15-id$^+$, they use different V(D)J recombinations and no single V(D)J recombination dominates within the MZB IgH repertoire (*Table 4*). In essence, the selection of a predominant V(D)J nucleotide sequence encoding a given CDR3 peptide is unique for the B-1a IgH repertoire.

## Multiple distinct V(D)J recombinations that encode the same CDR3 peptide in neonatal and young mice converge to a single identical V(D)J sequence in all adults

In 2–7 day animals, a few selected V(D)J nucleotide sequences, such as PtC-binding sequences, have already emerged as the predominant V(D)J recombination for their corresponding CDR3 peptide (*Figure 6A*, pattern II). However, most of the selected V(D)J nucleotide sequences, including T15Id$^+$, do not initially represent the predominant recombination for their corresponding CDR3 peptide. In particular, some CDR3 peptides are each encoded by multiple different V(D)J recombinations with similar frequencies in neonate mice. However, after weaning, a particular V(D)J recombination gradually increases its representation until it dominates in the adult B-1a IgH repertoire (*Figure 6A*, pattern I). In essence, although multiple distinctive V(D)J recombinations encoding the same CDR3 peptide exist in the neonatal/young B-1a IgH repertoire, a single identical V(D)J recombination sequence is selected to encode the particular CDR3 peptide in adult repertoire of almost all individuals.

In accordance with this finding, quantification of the diversity of V(D)J recombination events for each CDR3 peptide reveals the profound convergent recombination in the neonatal B-1a IgH repertoire. Thus, about 30% of CDR3 peptide sequences in splenic B-1a IgH repertoire at 2–6 day are encoded by more than one V(D)J recombination (entropy >0.5, *Figure 6B,C*), and about 10% of CDR3 peptide sequences show the highest level of convergent recombination (entropy >1.5, *Figure 6B,C*, the higher the entropy value, the more diverse the V(D)J recombinations). However, the frequency of CDR3 peptides showing convergent recombinations steadily decrease until the animals reach adulthood (2 months), after which very few (<1%) CDR3 peptide sequences show the multiple V(D)J recombinations (entropy >1.5, *Figure 6B,C*).

The step-wise decreases in the level of convergent recombination as animals age indicate the potent selection that over-time shapes the B-1a IgH repertoire. In most cases, the related V(D)J sequences that 'converge' to encode the same CDR3 peptide share the same D and J segments but use distinct V$_H$ genes (*Figure 6—figure supplement 1*). Therefore, despite encoding the same CDR3 peptide sequence, these related V(D)J sequences differ in their upstream regions including the CDR2 (*Figure 6—figure supplement 1*). These upstream differences, which can contribute to ligand binding, may be central to the selection of the predominant V(D)J sequence for the corresponding CDR3 peptide.

## AID-mediated SHM in pre-immune B-1a IgV$_H$ initiates after weaning and cumulatively increases the IgH repertoire diversity thereafter

Greater than 25% of splenic B-1a IgH sequences in 4–6 month old mice have at least one nucleotide change (*Figure 7A*). Such mutations are principally mediated by AID because they are rare (<2%) in splenic B-1a cells from age-matched AID-deficient mice (*Figure 7A*). The SHM even targets V(D)J sequences that are positively selected into the shared B-1a IgH repertoire in wild type mice (but not in AID-deficient mice) (*Figure 7B,D*). The observed mutations, most of which result in amino acid changes, are largely targeted AID hotspots, i.e., DGYW (D = A/G/T; Y = C/T; W = A/T) or WRCH (R = A/G, H = T/C/A) (*Di Noia and Neuberger, 2007*) (*Figure 7B,C*).

In contrast, mutations are minimal in IgV$_H$ of splenic FOB, MZB and peritoneal B-2 cells from adult mice (*Figure 7A*). Interestingly, the frequency of mutated IgH sequences in peritoneal B-1a cells in 4-6 month old mice is substantially lower than that in age-matched splenic B-1a cells and mutations are mainly single nucleotide change (*Figure 7A*).

SHM in splenic B-1a IgV$_H$ initiates after weaning and the frequency of mutated IgH transcripts increases with age. Thus, mutations are minimally detectable in the IgV$_H$ of splenic B-1a cells from

neonates (2–7 days) and young mice (2–3 weeks), are at lower frequencies in 2 month old mice, and are at substantially higher frequencies in 4–6 month old animals (*Figure 7A*). This age-dependent increase in splenic B-1a IgV$_H$ mutation argues that the detected SHM is not due to contamination with co-sorted B cells of other subsets, including GC cells, i.e., cells with the germinal center phenotype (GL7$^+$ CD38$^{lo}$ CD95$^{hi}$) are not detectable in the splenic B-1a population (*Figure 7—figure supplement 1*).

Furthermore, SHM is cumulative, becoming more pronounced with age. Thus, roughly 25% of IgH sequences from 4–6 month old splenic B-1a samples contain > = 1 nucleotide change, 19% contain > = 2 changes, and 9% contain > = 4 changes (*Figure 7A* and *Figure 7—figure supplement 2*). This translates to an average SHM rate of roughly 5 per 10$^3$ base pairs (bp) (*Figure 7E*), the similar range as that for SHM in GC responses, i.e., 10$^{-3}$ bp per generation (*Wagner and Neuberger, 1996*). Both the frequency of mutated sequences and the mutation rate for splenic B-1a samples from 2 month old mice are substantially lower than those in 4–6 month old mice (*Figure 7A,E*), further supporting that the SHM in the splenic B-1a IgV$_H$ is an accumulative process.

## Age-dependent progressive increase in the splenic B-1a IgV$_H$ mutations is accompanied by increased class-switching

Class switch recombination (CSR) is another genetic alteration process that somatically diversifies rearranged IgH genes. Both SHM and CSR are triggered by AID, which targets and introduces lesions in the IgV region for SHM and the switch regions for CSR (*Muramatsu et al., 2000*;

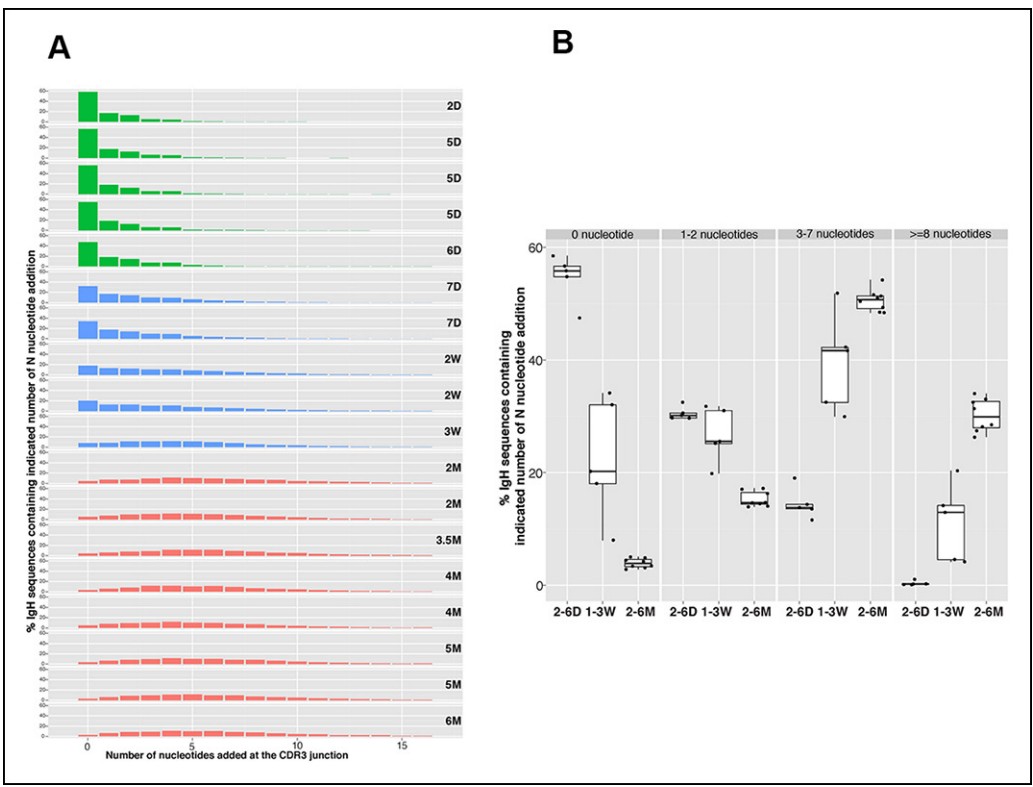

**Figure 4.** N nucleotide insertion distribution patterns for the B-1a pre-immune IgH repertoires during ontogeny. (A) Percentage of IgH sequences containing the indicated number of N nucleotide insertions at the IgH CDR3 junctions (V-DJ + D-J) is shown for each spleen B-1a sample from mice at indicated ages (shown at the right). To minimize the impact of self-renewal on the N-addition profile, normalized data are presented. Thus, each distinct IgH sequence containing indicated N nucleotide insertions is counted as one regardless how many times this sequence was detected. Note that the N insertion pattern changes as animals age. Colors distinguish three age-related patterns: green, D2 to D6; blue, D7 to 3W; red, 2M to 6M. (B) Percentages of IgH sequences containing the indicated N-nucleotide insertions (shown at the top) for splenic B-1a samples at the indicated ages are shown. Each dot represents data from an individual mouse, except for day 2 sample, n = 5-7.

*Chaudhuri and Alt, 2004*). Although both events require AID, SHM and CSR employ different enzymes and thus can occur independently (*Li et al., 2004*). Nevertheless, since they usually occur at the same differentiation stage and both are initiated by AID, the question arises as to whether the detected SHM in B-1a IgH is associated with CSR.

Our method allows detection of all different Ig isotypes. For each B cell sample, we quantified the frequency of IgH sequences expressing a given isotype and examined the relationship between the isotype profiles to the mutation status. Consistent with the close relationship between CSR and SHM, wild-type B cell samples that have minimal IgV_H mutations, including the splenic FOB, MZB, peritoneal B-2, neonate splenic B-1a (2–7 days), young splenic and peritoneal B-1a (2–3 weeks), rarely express class-switched transcripts (*Table 5*). Similarly, for B cell populations that show lower levels of mutation, e.g., splenic B-1a from 2 month old animals and peritoneal B-1a from 2–6 month old animals, the majority of both mutated and non-mutated sequences are either IgM or IgD and thus rarely class-switched (*Figure 8A*, *Table 5*).

In contrast, both the mutated and non-mutated IgH sequences from splenic B-1a in 4–6 month old animals contain class-switched Ig (*Figure 8*). Importantly, the class-switched Ig (mainly IgG3, IgG2b, IgG2c and IgA) represents a significantly higher proportion of the mutated sequences than of the non-mutated sequences (*Figure 8A*, *Table 6*), indicating that the increased SHM with age in the splenic B-1a IgH repertoire is accompanied by increased class-switching. However, despite the increased class switching among mutated sequences, the frequency of class-switched sequences appears not to correlate with the increased number of mutations (*Figure 8B*). Consistent with the class-switching dependence on AID, we did not detect isotypes other than IgM and IgD in splenic B-1a cells from 4–5 month old AID-deficient mice (*Table 5*).

The splenic B-1a cells that express class-switched Ig still express IgM on the surface, since cells were sorted as IgM^hi IgD^lo/- dump^- CD19^+ CD93^- CD21^-/lo CD23^- CD43^+ CD5^+. In addition, IgM^+

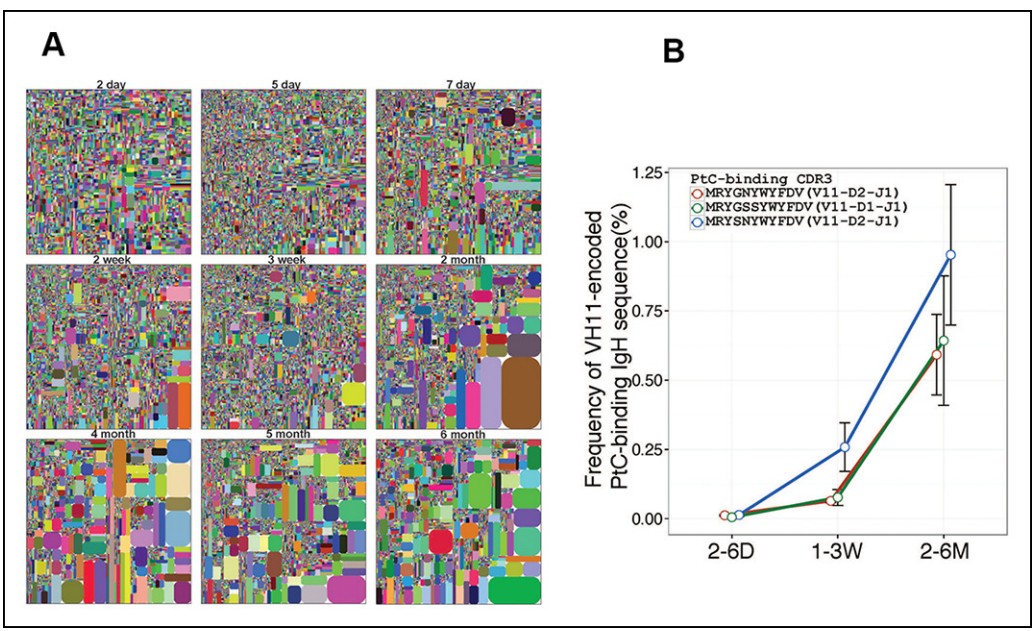

**Figure 5.** Certain V(D)J sequences increase progressively with age in the B-1a pre-immune IgH repertoire. (**A**) IgH CDR3 tree map plots for splenic B-1a samples from mice at different ages are shown. Each plot represents data for an individual mouse, except for the day 2 sample. Recurrent sequences are visualized as larger contiguously-colored rectangles in each plot. (**B**) Relative frequencies of three PtC-binding IgH CDR3 sequences in indicated splenic B-1a sample groups (n = 5–8 for each group) are plotted with mouse age. Sequence information (peptide and V(D)J recombination) is shown at the top.

The following figure supplement is available for figure 5:

**Figure supplement 1.** The peritoneal B-1a IgH repertoire is increasingly restricted during ontogeny.

**Table 4.** MZB IgH repertoires use different V(D)J recombination sequences to encode the same CDR3 peptide as that of B-1a anti-PC T15Id[+].

| MZB sample Id | Age (Months) | V(D)J recombination |
|---|---|---|
| 7630 | 2 | V1-76 D1-1 J1 and V1-39 D1-1 J1 |
| 10658 | 2 | V1-76 D1-1 J1 |
| 8700 | 4 | V1-72 D1-1 J1 and V8-12 D1-1 J1 |
| 8701 | 5 | V1-58 D1-1 J1 and V1-61 D1-1 J1 |
| 13338 | 4 | V1-61 D1-1 J1 and V5-16 D1-1 J1 |

Column 1: individual MZB samples tested; column 2: age of mouse for each MZB sample; column 3: for each MZB sample, V(D)J recombination events that encode ARDYYGSSYWYFDV, which is the CDR3 peptide associated with B-1a anti-PC T15Id[+].

cells described here barely co-express other surface isotypes. Thus the class-switched transcripts are derived from IgM[+] cells that apparently have already undergone class switching but have yet to turn off IgM surface protein expression. Since all of the cell preparation, staining and sorting were performed equivalently for all samples, our finding that the class-switched transcripts were selectively and predominantly detected in splenic B-1a cells from 4–6 month old mice argues that the detection of these transcripts is not due to contamination or other technical problems.

### The V(D)J selection and AID-mediated diversification that uniquely act in B-1a IgH repertoire operate comparably in germ-free and conventional mice

The microbiota are often thought to participate in shaping the repertoire of 'natural' antibodies, which is largely produced by B-1a (*Baumgarth et al., 2005*). Nevertheless, we find that germ-free (GF) animals have normal numbers of B-1a cells in spleen (*Figure 9—figure supplement 1*). Notably, the splenic B-1a IgH repertoires in age-matched (4-5 month old) specific pathogen free (SPF) and GF mice are very similar: 1) their IgH repertoires are comparably less diversified and enriched in the recurrent V(D)J sequences (*Figures 2A*, *9A*, *Table 7*); 2) their $V_H$ usage patterns show no significant differences (*Figure 3—figure supplement 1E*); 3) their CDR3 peptide expressions show a comparable extent of sharing between SPF and GF mice (*Figure 9B*); and 4) a substantial proportion of V(D)J sequences selected in the B-1a IgH repertoire in adult SPF mice are similarly selected in the B-1a IgH repertoire in GF mice (*Table 3*).

Further, hypermutation occurs equally in the splenic B-1a $IgV_H$ in 4–6 month old SPF and GF mice, i.e., the frequency of mutated sequences and the mutation rate are comparable under two conditions (*Figure 7A,E*). Indeed, AID introduces mutations into the identical V(D)J sequences expressed by splenic B-1a cells from either SPF or GF mice (*Figure 7B,C*). Finally, similar to SPF mice, AID-mediated class-switch occurs comparably in splenic B-1a cells from GF mice (*Figure 8A*). Since the V(D)J selection, hypermutation and class-switching operate comparably in splenic B-1a from GF and SPF mice, we conclude that the somatic mechanisms that select and diversify B-1a IgH repertoire over time are not driven by microbiota-derived antigens.

Nevertheless, the environment has a strong impact on the isotype representation. IgA transcripts are readily detected in splenic B-1a from 4–6 month old SPF mice; however, these transcripts are minimally detected in the splenic B-1a from age-matched GF mice (*Figure 8B*, *Table 6*). This finding is consistent with the recognition that class-switching to IgA is strongly associated with the presence of gut microbiota (*Kroese et al., 1989*; *Macpherson et al., 2000*).

### Discussion

Studies presented here open a new perspective on the origin and breadth of humoral immunity that protect against invading pathogens and regulate autoimmunity. Recent studies have already shown B-1a develops prior to and independent from BM HSC, which fail to generate B-1a but fully constitute FOB and MZB compartment (*Ghosn et al., 2012*; *Yoshimoto et al., 2011*). Cohering the

fundamental difference in their development origin, our studies reveal two distinct IgH repertoires that develop at different times and are shaped by distinct functional mechanisms.

The first of these repertoires is expressed in B-1a cells. The *de novo* IgH rearrangements in this repertoire occur mainly during the first few weeks of age and largely cease thereafter. Then B-1a cells persist as a self-replenishing population. The B-1a repertoire, however, continues to evolve under stringent selection. Thus, certain V(D)J sequences increase with age, and certain V(D)J nucleotide sequences gradually emerge as the predominant recombinations encoding the specific CDR3 peptides in all adults. Furthermore, the age-dependent V(D)J selection coincides with the progressive introduction of IgV$_H$ mutation and increased class-switch. Importantly, the V(D)J selection and AID-mediated diversification occur comparably in *germ-free* and conventional mice, indicating that these unique repertoire-defining mechanisms are not driven by microbiota-derived antigens.

In contrast, MZB, FOB and peritoneal B-2 cells develop later, and continuously develop *de novo* from BM HSC throughout life and express drastically different IgH repertoire(s). Their IgH repertoires tend to preferentially utilize V1 (J558) family, are far more diverse and less repetitive and, unlike B-1a cells, show no apparent selection for particular V(D)J recombination sequences and do not show IgV$_H$ mutation and class-switch. In essence, AID introduces SHM and CSR in these B cell subsets only when they respond to their cognate antigens that are largely exogenous in nature.

These findings were enabled by employing the amplicon-rescued multiplex PCR technology, which allows the capture and amplification of Ig transcripts from a given B cell population in an inclusive and quantitative fashion. Specifically, the first RT-PCR reaction, which uses an array of gene-specific primers for almost all V$_H$ families and all constant (C$_H$) genes, is carried out only for a few cycles. The second round of PCR is then carried out with communal primers that recognize the unique sequences tagged into each of the V$_H$ and C$_H$ primers. Since these 'tag sequences' were already introduced during the initial cycles, the use of the communal primers assures that all of the targets are amplified with reduced bias during the following exponential phase of amplification. Coupled with the next generation sequencing, our method is quite robust and allows detection of diverse Ig transcripts that collectively carry about 100 V$_H$ genes associated with different isotypes.

As with other bulk RNA sequence measurement, our methods cannot determine the absolute number of each Ig transcript in a given B cell population. Hence the actual number of cells expressing a certain Ig sequence is unknown. In addition, our methods do not allow determination of whether certain sequences associated with distinct isotypes belong to the same cell. Further, since the Ig transcript copy number variation among cells is unknown, the frequency of a given Ig transcript is roughly viewed as the relative index of the frequency of cells expressing this Ig transcript. This assumption is generally valid since our studies exclude plasmablast and plasma cells, which do not express surface CD5. Since B-1a cells are well-known to undergo self-replenishment in adult, the dramatic increase in certain V(D)J sequences in the B-1a IgH repertoire over time likely reflects the expansion of cells expressing this particular V(D)J sequence.

Single cell sequencing analysis has advantages in reducing technical bias and in enabling paired IgH/IgL sequencing. Nevertheless, sequencing costs are still a big hurdle to the large-scale single cell analysis, which, as our studies demonstrate, is necessary to develop a comprehensive view of the various B cell subset repertoires. Therefore, at least for the present, our approaches are more efficient and practical.

B-1a produce 'natural' antibodies, many of which recognize endogenous (self) antigens (*Baumgarth et al., 2005*) and play house-keeping roles in clearing the cellular debris or metabolic wastes (*Shaw et al., 2000*; *Binder and Silverman, 2005*). Since the natural antibodies can also react/cross-react with microorganism-derived antigens, they also participate in the first line of immune defense (*Ochsenbein et al., 1999*; *Baumgarth et al., 2000*). Germ-free mice have normal levels of circulating 'natural' IgM (*Bos et al., 1989*). Earlier immunologists have postulated that the natural antibody repertoire is selected by endogenous (self) antigens (*Jerne, 1971*; *Coutinho et al., 1995*). Our studies, which demonstrate that B-1a IgH repertoire (hence the re-activities of natural antibodies) is highly similar between individual adult C57BL/6 mice, regardless of whether the animals are reared in conventional or germ-free facilities, introduce the solid evidence supporting this argument.

Our studies also demonstrate that the B-1a IgH repertoire is selected over time. Thus, recurrent V(D)J sequences appear later, and most of the V(D)J sequences that are selected to be conserved in all individuals do not emerge until the animals reach the adulthood. As a result, the sequence

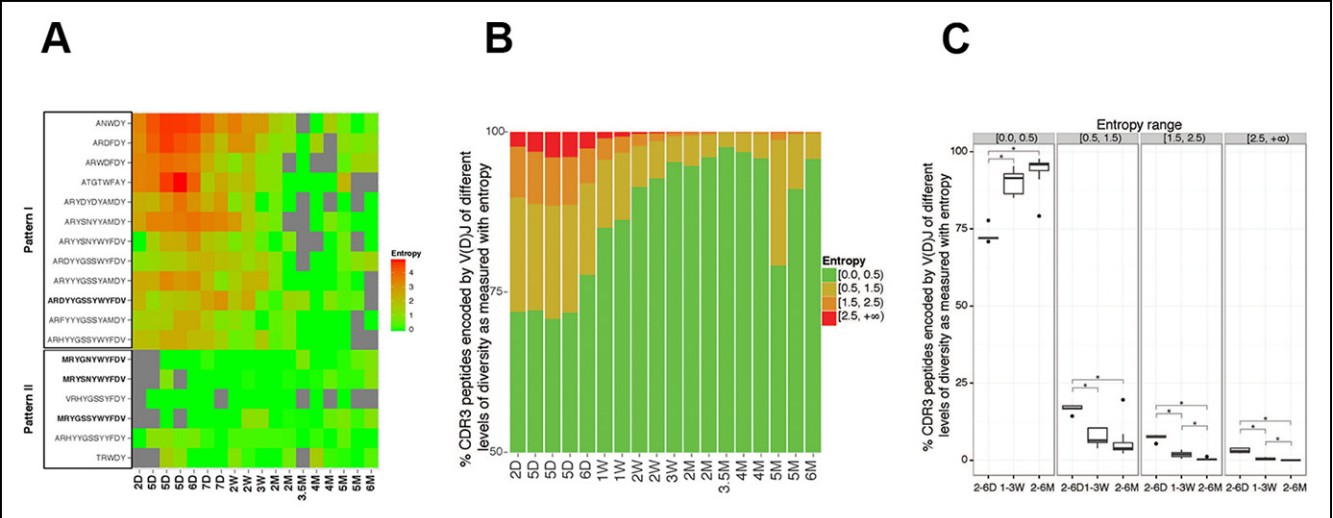

**Figure 6.** The level of convergent recombination in the B-1a IgH repertoire declines with age. (**A**) Entropy heat map showing the diversity of V(D)J recombination events for each indicated CDR3 peptide (shown at the left) in splenic B-1a samples at different ages (shown at the bottom). The higher the entropy value, the more diverse the V(D)J recombinations for a given CDR3 peptide. CDR3 peptide sequences for T15 Id$^+$ anti-PC (pattern I) and anti-PtC (pattern II) antibody are shown in bold. (**B**) The diversities of the V(D)J recombination for each CDR3 peptide for the indicated splenic B-1a samples (shown at the bottom) are quantified as entropy values (see Methods and materials), which are ranked into 4 ranges (shown at the right). For each sample, the frequencies of CDR3 peptide sequences belonging to each entropy range are shown as stacks. (**C**) Splenic B-1a samples are grouped based on age. For each group (n = 5–7), the frequencies of CDR3 peptide sequences belonging to each of four entropy ranges are shown. *p<0.05, Welch's t-test.

The following figure supplement is available for figure 6:

**Figure supplement 1.** Distinct V(D)J sequences encoding the same CDR3 peptide differ in $V_H$ usage.

composition of B-1a IgH repertoire in adult mice becomes much less random than that expressed in neonate and younger mice. Furthermore, the convergent selection of a particular V(D)J recombination sequence encoding a specific CDR3 peptide indicates that the selection is strikingly precise and occurs at both the protein and the nucleotide level.

Unexpectedly, our studies find that both SHM and CSR participate in diversifying the B-1a IgH repertoire. However, unlike GC response SHM, which occurs within a few days following antigenic stimulation, SHM in B-1a IgV$_H$ starts after weaning and is cumulative with age. The progressive increase in the SHM is also associated with increased class switching. Most importantly, SHM and CRS occur comparably in *germ-free* and conventional mice, indicating that SHM and CSR in the B-1a primary IgH repertoire are not driven by microbiota-derived antigen. Since B-1a cells are well-known to produce anti-self antibodies, stimulation by endogenous antigens is likely the major driving force for the AID-mediated diversification processes.

Ongoing SHM in the absence of external antigens influence have been reported in sheep B cells (*Reynaud et al., 1995*). The accumulation of SHM in B-1a IgV$_H$ over time likely represents a similar strategy to further diversify their restricted Ig repertoire as animal age. Such diversification may potentiate defenses against newly encountered pathogens. However, the age-dependent accumulative SHM, which is likely driven by self-antigens, may also increase the risk of autoimmune disease due to pathogenic high affinity auto-reactive antibodies. Indeed, deregulated B-1a growth have been reported in NZB/W mice, where autoantibody-associated autoimmune disease develops as animal age (*Hayakawa et al., 1984*).

AID-mediated mutagenesis in B-1a IgV$_H$ may occasionally introduce mutations elsewhere in the genome that facilitate dysregulated growth and neoplastic transformation, e.g., B-chronic lymphocytic leukemia (B-CLL) (*Stall et al., 1988*; *Kipps et al., 1992*; *Phillips and Raveche, 1992*). Although the mechanism by which the IgM$^+$ splenic B-1a cells from older mice express class-switched Ig transcripts remains elusive, this finding suggests that certain cells are undergoing vigorous genetic

**Table 5.** B cell samples that show minimal or low level mutations in IgV$_H$ rarely express class-switched transcripts.

| Sample Id | subset | age | strain | non-mutated or mutated sequences (%) | | IgM (%) | IgD (%) | IgG1(%) | IgG3 (%) | IgG2c (%) | IgG2b (%) | IgE (%) | IgA (%) |
|---|---|---|---|---|---|---|---|---|---|---|---|---|---|
| 13965 | sB-1a | 4M | AIDKO | non-mutated | 98.2 | 99.5 | 0.5 | | | | | | |
| | | | | mutated | 1.8 | 100 | | | | | | | |
| 13968 | sB-1a | 5M | AIDKO | non-mutated | 99 | 100 | | | | | | | |
| | | | | mutated | 1 | 100 | | | | | | | |
| 13971 | sB-1a | 4M | AIDKO | non-mutated | 98.4 | 99.6 | 0.4 | | | | | | |
| | | | | mutated | 1.6 | 100 | | | | | | | |
| 13972 | sB-1a | 4M | AIDKO | non-mutated | 100 | 99.9 | 0.1 | | | | | | |
| 8704 | pB-1a | 4M | AIDKO | non-mutated | 98.2 | 99.9 | 0.1 | | | | | | |
| | | | | mutated | 1.8 | 100 | | | | | | | |
| 13973 | pB-2 | 5M | AIDKO | non-mutated | 100 | 91.6 | 8.4 | | | | | | |
| 8700 | MZB | 5M | WT | non-mutated | 99.8 | 99.9 | 0.1 | | | | | | |
| 8701 | MZB | 4M | WT | non-mutated | 98.6 | 99.9 | 0.1 | | | | | | |
| 7630 | MZB | 2M | WT | non-mutated | 99.5 | 99.9 | 0.1 | | | | | | |
| 10658 | MZB | 2M | WT | non-mutated | 100 | 100 | | | | | | | |
| 8702 | FOB | 5M | WT | non-mutated | 99.8 | 98.6 | 1.4 | | | | | | |
| 13966 | FOB | 3.5M | WT | non-mutated | 99.5 | 99.7 | 0.3 | | | | | | |
| 7631 | FOB | 2M | WT | non-mutated | 99.3 | 72.9 | 27.1 | | | | | | |
| 7629 | pB-2 | 4M | WT | non-mutated | 98.4 | 88.4 | 11.5 | | | | | | |
| | | | | mutated | 1.6 | 89 | 11 | | | | | | |
| 13969 | pB-2 | 3.5M | WT | non-mutated | 99.5 | 90.1 | 9.9 | | | | | | |
| 13974 | sB-1a | day 2 | WT | non-mutated | 99.2 | 99.8 | 0.2 | | | | | | |
| 13000 | sB-1a | day 2 | WT | non-mutated | 99.2 | 100 | | | | | | | |
| 10659 | sB-1a | day 5 | WT | non-mutated | 99.2 | 100 | | | | | | | |
| 9866 | sB-1a | day 5 | WT | non-mutated | 100 | 100 | | | | | | | |
| 10651 | sB-1a | day 5 | WT | non-mutated | 99.7 | 100 | | | | | | | |
| 10652 | sB-1a | day 6 | WT | non-mutated | 99.4 | 100 | | | | | | | |
| 9868 | sB-1a | day 7 | WT | non-mutated | 99.3 | 99.9 | 0.1 | | | | | | |
| 9865 | sB-1a | day 7 | WT | non-mutated | 99.5 | 99.6 | 0.4 | | | | | | |
| 11168 | sB-1a | 2W | WT | non-mutated | 99.1 | 99.9 | 0.1 | | | | | | |
| 13005 | sB-1a | 2W | WT | non-mutated | 99.5 | 100 | | | | | | | |
| 10654 | sB-1a | 3W | WT | non-mutated | 95.8 | 100 | | | | | | | |
| | | | | mutated | 4.2 | 100 | | | | | | | |
| 11160 | pB-1a | 2W | WT | non-mutated | 99 | 100 | | | | | | | |
| 10655 | pB-1a | 3W | WT | non-mutated | 99.2 | 100 | | | | | | | |
| 11163 | pB-1a | 1M | WT | non-mutated | 99.1 | 99.9 | | | | | | | |
| 7632 | sB-1a | 2M | WT | non-mutated | 88.1 | 99.1 | 0.9 | | | | | | |
| | | | | mutated | 11.9 | 99.9 | | | 0.1 | | | | |
| 10656 | sB-1a | 2M | WT | non-mutated | 88.1 | 99.8 | 0.2 | | | | | | |
| | | | | mutated | 11.9 | 100 | | | | | | | |
| 13004 | sB-1a | 2M | WT | non-mutated | 97.7 | 99.9 | | | | | | | |
| | | | | mutated | 2.3 | 94 | | | | | | | 6 |
| 13018 | pB-1a | 2M | WT | non-mutated | 91.8 | 100 | | | | | | | |
| | | | | mutated | 8 | 100 | | | | | | | |

*Table 5 continued on next page*

*Table 5 continued*

| Sample Id | subset | age | strain | non-mutated or mutated sequences (%) | | IgM (%) | IgD (%) | IgG1(%) | IgG3 (%) | IgG2c (%) | IgG2b (%) | IgE (%) | IgA (%) |
|---|---|---|---|---|---|---|---|---|---|---|---|---|---|
| 13660 | pB-1a | 2M | WT | non-mutated | 92.2 | 100 | | | | | | | |
| | | | | mutated | 7.6 | 100 | | | | | | | |
| 7628 | pB-1a | 2M | WT | non-mutated | 92.3 | 99.5 | 0.4 | | 0.1 | | | | |
| | | | | mutated | 7.1 | 99.6 | 0.2 | | 0.2 | | | | |
| 8705 | pB-1a | 4M | WT | non-mutated | 93.8 | 99.7 | 0.3 | | | | | | |
| | | | | mutated | 4.4 | 99.9 | | | | | | | |
| 9870 | pB-1a | 4M | WT | non-mutated | 86.4 | 99.9 | | | | | | | |
| | | | | mutated | 12.6 | 99.9 | | | | | | | |
| 11165 | pB-1a | 5M | WT | non-mutated | 98.1 | 99.9 | | | | | | | |
| | | | | mutated | 1.5 | 100 | | | | | | | |
| 8707 | pB-1a | 5M | WT | non-mutated | 91.6 | 97.2 | 0.1 | | 2 | 0.5 | 0.1 | | 0.1 |
| | | | | mutated | 6.2 | 97.9 | 0.1 | | 2 | | | | |
| 9861 | pB-1a | 6M | WT | non-mutated | 82.4 | 99.6 | 0.4 | | | | | | |
| | | | | mutated | 17.5 | 100 | | | | | | | |

Table lists each individual B cell sample (labeled as distinct Id number) from wild-type (WT) or AID-deficient (AIDKO) mice. The mouse age and sample subset information are also shown. For each sample, the sequences are divided into non-mutated or mutated (> = 1 nucleotide change) categories, the frequencies of each category are shown. For each category, the frequencies of sequences with each isotype are also shown.

alteration that may share the similar mechanisms that underlie the malignant transformation. In fact, cells with simultaneous expression IgM and class-switched Ig transcripts have been reported in B-CLL and other B cell tumors (*Oppezzo et al., 2002*; *Kinashi et al., 1987*).

The splenic and peritoneal B-1a IgH repertoires show similar characteristics. Both repertoires become more restricted with age with increased recurrent V(D)J sequences (*Figure 5—figure supplement 1*) and retain the positive selected V(D)J sequences in adult animals. However, our studies reveal the key repertoire differences between B-1a cells at their two native locations. Although both repertoires show extensive CDR3 sharing among individual mice, the peritoneal B-1a IgH repertoire is more similar to neonatal splenic B-1a repertoire and shows a significantly higher level of CDR3 peptide sharing among individual mice than the splenic B-1a repertoire (*Figure 2B*). In addition, the peritoneal B-1a IgH repertoire is more biased in using V6-6 (J606), V9-3 (Vgam3.8), V2-9 (Q52) and V2-6-8 (Q52), which are preferentially expressed in splenic B-1a from neonate and younger mice.

These findings suggest that peritoneal B-1a cells are enriched for cells that are generated during neonatal and young age of life, thus are largely consist of cells migrated from spleen into PerC when the animals were younger. This argument is further supported by the findings that the frequencies of mutated sequences in the peritoneal B-1a cells from 4-6 month old mice are substantially lower and the mutations are mainly single nucleotide changes whereas a proportion of IgH sequences with multiple mutations is detected in splenic B-1a cells from the same aged mice (*Figure 7*).

MZB and B-1a share many phenotypic and functional characteristics (*Martin and Kearney, 2001*). Our studies show that the MZB IgH repertoire differs drastically from the B-1a IgH repertoire, but is very similar to the repertoires expressed by splenic FOB and peritoneal B-2. Since MZB and FOB cells are mainly derived from BM HSC (*Ghosn et al., 2012*), there findings collectively support the idea that these B cells belong to the same (i.e.,B-2) developmental lineage. Nevertheless, the MZB repertoires from individual mice contain substantially higher levels of common CDR3 sequences (peptides) than the splenic FOB and peritoneal B-2 repertoires (*Figure 2B*).

Years ago, we postulated that B-1a and B-2 B cells belong to distinct developmental lineages that are evolved sequentially to play complementary roles in immunity (*Herzenberg and Herzenberg, 1989*). The sequence data presented here, which reveal the key distinctions in the repertoires as well as the repertoire-defining mechanisms between B-1a and B-2 subsets, support this argument

and greatly extend our earlier version. These key distinctions provide the genetic bases for their well-known fundamental functional difference between B-1a and other B subsets. In particular, they are central to vaccine development, where the recognition that the B cells have distinct targeting antibody repertoires clearly invites attention. In addition, our findings offer insights in understanding the origins and behaviors of B cell neoplasms, particularly B-CLL, and the autoimmune diseases in which over production of autoantibodies is implicated in the pathology.

## Materials and methods

### Mice

C57BL/6J mice were purchased from the Jackson Laboratory. AID-deficient C57BL6/J mice were kindly provided by Dr. Michel Nussenzweig (Rockefeller University). Mice were breed and kept in the

**Table 6.** Both the mutated and non-mutated IgH sequences obtained from splenic B-1a cells in 4-6 month old animals contain class-switched Ig.

| sample Id | subset | age | condition | non-mutated or mutated sequences (%) | | IgM (%) | IgD (%) | IgG1 (%) | IgG3 (%) | IgG2c (%) | IgG2b (%) | IgE (%) | IgA (%) |
|---|---|---|---|---|---|---|---|---|---|---|---|---|---|
| 9867 | sB-1a | 6M | SPF | non-mutated | 50 | 95.8 | | | 2.7 | 0.6 | 0.8 | | 0.1 |
| | | | | mutated | 50 | 65 | | 0.03 | 16.8 | 4.3 | 4.8 | | 9.1 |
| 8699 | sB-1a | 4M | SPF | non-mutated | 56 | 99.5 | | | 0.3 | 0.1 | 0.1 | | |
| | | | | mutated | 44 | 89.9 | | | 3.4 | 2 | 1.3 | | 3.4 |
| 9863 | sB-1a | 4M | SPF | non-mutated | 74.1 | 93.9 | 0.2 | | 3.6 | 1 | 1.3 | | |
| | | | | mutated | 25.9 | 44.2 | | | 41 | 9.7 | 3.7 | | 1.4 |
| 13970 | sB-1a | 3.5M | SPF | non-mutated | 74.1 | 92.7 | 0.5 | | 3.7 | 1.7 | 1.3 | | |
| | | | | mutated | 25.9 | 92.1 | | | 2.5 | 0.5 | 4.8 | | 0.1 |
| 13342 | sB-1a | 4M | SPF | non-mutated | 88.9 | 97.6 | 0.5 | | 0.8 | 0.6 | 0.2 | | 0.3 |
| | | | | mutated | 11.1 | 85.2 | | | 0.3 | 0.2 | 7.3 | | 7 |
| 13337 | sB-1a | 4M | GF | non-mutated | 69.7 | 98.5 | 0.1 | | 0.8 | 0.4 | 0.1 | | |
| | | | | mutated | 30.3 | 79 | | | 15.3 | 5 | 0.7 | | |
| 13003 | sB-1a | 4M | GF | non-mutated | 74.8 | 97.2 | 0.3 | 0.2 | 0.5 | 0.2 | 1.6 | | |
| | | | | mutated | 25.2 | 89.8 | | 1.1 | 2.3 | 1.2 | 5.6 | | |
| 13341 | sB-1a | 4M | GF | non-mutated | 78.2 | 99 | 0.1 | | 0.2 | 0.1 | 0.6 | | |
| | | | | mutated | 21.8 | 72.2 | | | 9.6 | 5.5 | 12.6 | | 0.1 |
| 13017 | sB-1a | 4M | GF | non-mutated | 80.9 | 95.6 | 0.4 | | 2 | 1 | 1 | | |
| | | | | mutated | 19.1 | 79 | 0.2 | | 7.9 | 3.9 | 8.9 | | 0.1 |
| 13002 | sB-1a | 4M | GF | non-mutated | 88.5 | 97.4 | 0.5 | | 0.6 | 0.2 | 1.3 | | |
| | | | | mutated | 11.5 | 63.8 | | | 14.8 | 8.4 | 13 | | |

Table lists individual splenic B-1a cell sample sorted from 4-6 month old C57BL6/J mice reared under either specific pathogen free (SPF) or germ-free (GF) condition. For each sample, the sequences are divided into non-mutated or mutated (> = 1 nucleotide change) categories, the frequencies of each category are shown. For each category, the frequencies of sequences expressing each isotype are shown. The data for germ-free animals is discussed at the end of the result section.

**Table 7.** Top 10 highly recurring CDR3 sequences (peptide and V(D)J recombination) detected in listed splenic B-1a samples from age-matched SPF and GF mice.

| sB-1a samples (4 months) | Top 10 IgH CDR3 sequences | | |
|---|---|---|---|
| | | Peptide | V(D)J |
| germ-free #1 | 1 | MRYGSSYWYFDV | V11-2 D1-1 J1 |
| | 2 | ARGAY | V1-80 J2 |
| | 3 | ARNPDGYYTYYYAMDY | V2-2 D2-9 J4 |
| | 4 | ARDPFYYYGSSYWYFDV | V5-16 D1-1J1 |
| | 5 | MRYSNYWYFDV | V11-2 D2-6 J1 |
| | 6 | AITRAY | V1-55 J3 |
| | 7 | ARRYYGSSYWYFDV | V1-55 D1-1 J1 |
| | 8 | ARSDYYGSSSLSY | V1-26 D1-1 J2 |
| | 9 | ASGGNYFDY | V1-75 J2 |
| | 10 | ARSLYN | V1-9 J2 |
| germ-free #2 | 1 | ARNYGSSYDY | V1-53 D1-1 J2 |
| | 2 | TRPSYYGSDY | V14-4 D1-1 J2 |
| | 3 | TRESYDGYYVWYAMDY | V5-9-1 D2-9 J4 |
| | 4 | ARGDY | V14-3 J2 |
| | 5 | ASNWAY | V1-53 D4-1 J2 |
| | 6 | MRYSNYWYFDV | V11-2 D2-6 J1 |
| | 7 | AKGDYYGSSYYFDY | V1-9 D1-1 J2 |
| | 8 | VRHGPRAFDY | V10-1 D3-2 J2 |
| | 9 | ARLNGDY | V1-69 J2 |
| | 10 | MRYGNYWYFDV | V11-2 D2-8 J1 |
| specific pathogen free #1 (from Caltech) | 1 | ASYSNSDV | V3-6 D2-6 J1 |
| | 2 | ARVSYSRAMDY | V14-3 D2-6 J4 |
| | 3 | ARSGNYGAMDY | V1-7 D2-8 J4 |
| | 4 | ASRLRSTFAY | V2-6-8 D1-1 J3 |
| | 5 | ARVTTVHAMDY | V1-55 D1-1 J4 |
| | 6 | ARNYGSSYWYFDV | V1-53 D1-1 J1 |
| | 7 | ARTPNWEARDY | V1-55 D4-1 J4 |
| | 8 | ARRYYGSSYWYFDV | V1-55 D1-1 J1 |
| | 9 | ARPLLYRYYFDY | V1-75 D2-6 J2 |
| | 10 | ARNYGSSYDWYFDV | V1-9 D1-1 J1 |
| specific pathogen free #2 (from Caltech) | 1 | ARGGIYYDYDEVYYYAMDY | V1-55 D2-4 J4 |
| | 2 | MRYSNYWYFDV | V11-2 D2-6 J1 |
| | 3 | ARDYYGSSWYFDV | V1-26 D1-1 J1 |
| | 4 | MRYGNYWYFDV | V11-2 D2-8 J1 |
| | 5 | MRYGSSYWYFDV | V11-2 D1-1 J1 |
| | 6 | ARYYDGYYGYYAMDY | V1-26 D2-4 J4 |
| | 7 | ALITTWYFDV | V1-78 D1-2 J1 |
| | 8 | ARHYYGSSWGY | V1-53 D1-1 J2 |
| | 9 | ARSFSPYYFDY | V1-26 J2 |
| | 10 | ARSHGYYPFDY | V1-54 D2-9 J2 |

*Table 7 continued on next page*

*Table 7 continued*

| sB-1a samples (4 months) | Top 10 IgH CDR3 sequences | | |
| --- | --- | --- | --- |
| | | Peptide | V(D)J |
| specific pathogen free #1 (from Stanford) | 1 | ARSADYGGYFDV | V1-64 D2-4 J1 |
| | 2 | ARGAY | V1-80 J2 |
| | 3 | ARSYYDYPWFAY | V1-76 D2-4 J3 |
| | 4 | ARRWLLNAMDY | V1-9 D2-9 J4 |
| | 5 | ARPYYYGSSPWFAY | V1-69 D1-1 J3 |
| | 6 | ARNDYPYWYFDV | V1-4 D2-4 J1 |
| | 7 | ARSGDY | V1-64 J2 |
| | 8 | ARVIGDY | V1-53 D2-14 J4 |
| | 9 | ARANY | V1-55 J3 |
| | 10 | AVNWDYAMDY | V1-84 D4-1 J4 |
| specific pathogen free #2 (from Stanford) | 1 | ARGNY | V1-80 J2 |
| | 2 | ARWVYYGSSSYWYFDV | V1-54 D1-1 J1 |
| | 3 | ARSSNYAMDY | V1-78 D2-11 J4 |
| | 4 | ARYYYGSNYAMDY | V7-3 D1-1 J4 |
| | 5 | ARGAY | V1-80 J2 |
| | 6 | ARRYYGSSYWYFDV | V1-55 D1-1 J1 |
| | 7 | ARSPYYSNYEGYFDV | V1-72 D2-6 J1 |
| | 8 | ARKNYGSSYWYFDV | V1-55 D1-1 J1 |
| | 9 | ARLEIYYGNYGRVFDV | V1-80 D2-8 J2 |
| | 10 | ARRDYYGSSYVLAY | V1-9 D1-1 J3 |

Table lists the top 10 highly recurring CDR3 sequences (peptide and V(D)J recombination) shown in each of CDR3 tree-map plot (*Figure 9A*).

Herzenberg laboratory colony under SPF conditions at the Stanford Veterinary Service Center (VSC). Spleens from germ-free C57BL6/J mice were provided by Dr. Sarkis Mazmanian (Caltech). Germ-free mice were maintained in sterile Trexler isolators and fed autoclaved food and water. Germ-free status was assayed monthly by aerobic and anaerobic plating; and by 16s rRNA PCR. Study protocols were approved by the Stanford VSC.

## Hi-dimensional FACS sorting

FACS staining has been previously described (*Yang et al., 2012*). Briefly, cell suspensions were incubated with LIVE/DEAD Aqua (Life Technologies, San Diego, CA), washed, and incubated with unconjugated anti-CD16/CD32 (FcRII/III) mAb to block Fc-receptors. Cells were then stained on ice for 20 min. with a 'cocktail' of fluorochrome-conjugated antibodies including: anti-CD21-FITC (Becton Dickenson, San Jose, CA), anti-CD43-PE (BD), anti-CD5-PE-Cy5 (BD), anti-CD19-PE-Cy5.5 (Life Technologies), anti-CD93 (AA41)-PE-Cy7 (eBioscience, San Diego, CA), anti-B220-APC (BD), anti-IgM-Alexa700 (Herzenberg lab), anti-IgD-APC-Cy7 (BioLegend, San Diego, CA), anti-CD23-Biotin (BD), anti-CD11b-PB (Life Technologies), anti-Gr-1-PB (Life Technologies), anti-TCRαβ-PB(Life Technologies), anti-CD11c-PB (Life Technologies), and anti-CD3?-PB (Life Technologies). After washing, cells were stained with Streptavidin-Qdot 605 (Life Technologies). Cells were sorted on FACS Aria (BD) at the Stanford Shared FACS Facility. Sorting purity was greater than 99%. Five types of B cell populations were sorted based on tissue and phenotype: splenic and peritoneal B-1a cells (dump⁻ CD19⁺ CD93⁻ IgM^hi IgD^-/lo CD21^-/lo CD23⁻ CD43⁺ CD5⁺); splenic FOB and peritoneal B-2 cells (dump⁻ CD19⁺ CD93⁻ IgM^lo IgD^hi CD23⁺ CD43⁻ CD5⁻); splenic MZB cells (dump⁻ CD19⁺ CD93⁻ IgM^hi IgD^-/lo

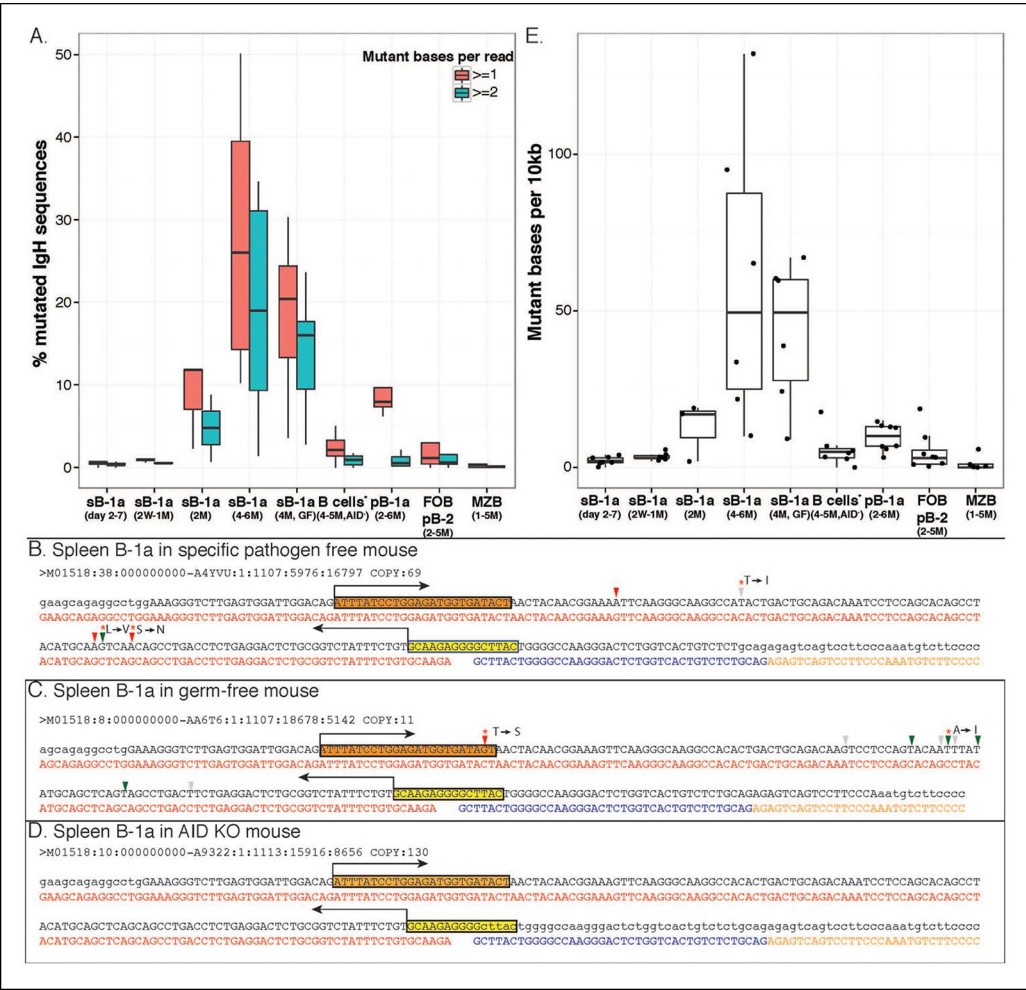

**Figure 7.** AID-mediated SHM accumulates on splenic B-1a IgV$_H$ with age. (**A**) Percentages of sequences containing > = 1 (red) or > = 2 (green) nucleotide changes for B cell samples from mice at the indicated ages are shown (n = 3-8). Seven B cell samples from 4-5 month old AID knockout mice include sB-1a (n = 4), pB-1a (n = 1), FOB (n = 1) and pB-2 (n = 1). Sequences with the identical V(D)J recombination encoding ARGAY CDR3 peptide obtained from splenic B-1a sample from 4 month old specific pathogen free mouse, (**B**) germ-free mouse (**C**) and AID knockout mouse (**D**) are listed. The nucleotide substitution is analyzed at the V$_H$ region stretching from the start of CDR2 (red box) to the beginning of CDR3 (yellow box). Obtained sequence (upper line) is aligned with the reference (lower line) for V1-80 (red), J3 (blue) and constant region of IgM isotype (orange). Mutations are highlighted with triangles; asterisks indicate mutations resulting in an amino acid change; red and blue triangles denote mutations in DGYW and WRCH motifs, respectively. (**E**) Numbers of mutations per 10$^4$ base pairs for indicated B cell group are shown. Each dot represents data from an individual sample (n = 3–8). The data for germ-free (GF) animals is discussed at the end of the Result section. Note: The mutation profiles for the splenic B-1a IgH libraries prepared by using either old (V$_H$12-3 deficient) or new primer set (V$_H$12-3 included) are highly similar (*Figure 7—figure supplement 3*).

The following figure supplements are available for figure 7:

**Figure supplement 1.** Splenic B-1a cells do not contain cells expressing GC phenotype.

**Figure supplement 2.** Percentage of sequences containing > = 4 nucleotides changes for each B cell group.

**Figure supplement 3.** Identical V(D)J recombination sequences containing identical mutated nucleotides are detected in sequence data sets for IgH libraries obtained by using either old or new primer set.

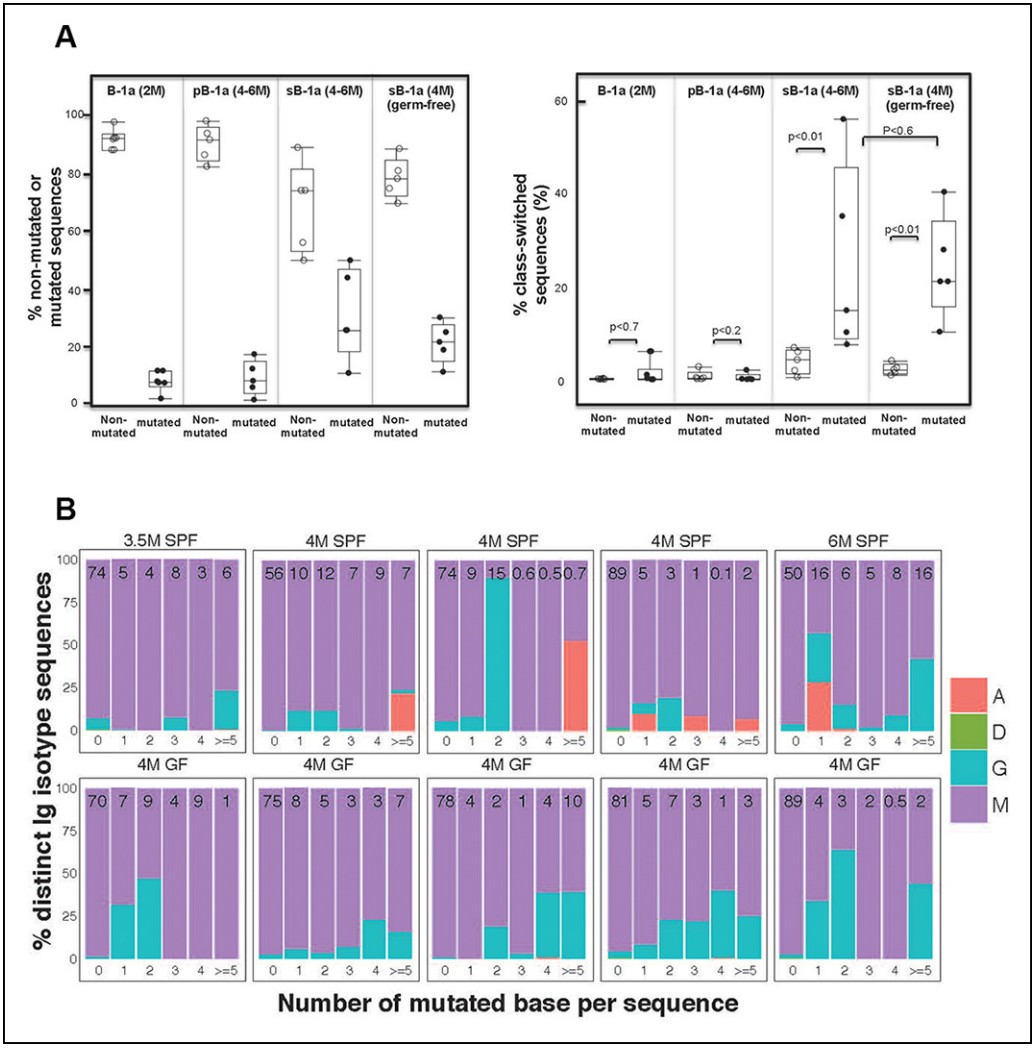

**Figure 8.** Progressive increase in the splenic B-1a IgV$_H$ mutation frequency with age is accompanied by increased class-switching. (**A**) *Left panel:* The frequencies of non-mutated or mutated (> = 1 nucleotide substitution) IgH sequences obtained from indicated B cell samples are shown; *Right panel:* The frequencies of sequences expressing class-switched isotypes (neither IgM nor IgD) among non-mutated or mutated sequences are shown. Each dot represents data from an individual sample (n = 5–6). p values are calculated based on the Nonparametric Wilcoxon test. (**B**) In each plot, the IgH sequences obtained from each splenic B-1a sample from 3.5–6 month old mice are divided into five categories, based on the number of mutated nucleotides (0, 1, 2, 3, 4, > = 5) per read. In each plot, the values shown at the top are the frequencies of sequences in each category. For each category of sequences, frequencies of the distinct isotype sequences are shown as stacks. A = IgA; D = IgD; G = IgG1 + IgG3 + IgG2c + IgG2b. Each plot represents the data for a splenic B-1a sample from an individual mouse reared under either specific pathogen free (SPF) (*upper plots*) or germ-free (GF) (*lower plots*) conditions. The data for germ-free (GF) animals is discussed at the end of the Result section.

CD21$^{hi}$ CD23$^{lo/-}$ CD43$^-$ CD5$^-$). 1-2 x 10$^4$ cells for each cell population were sorted directly into 0.5 mL Trizol LS (Life Technologies).

## Amplicon rescued multiplex PCR

RNA was extracted according to the protocol provided by Trizol LS (Life Technologies). RT-PCR reactions were conducted using a set of sequence specific primers covering almost all of mouse V$_H$ genes (forward primers) and constant C$_H$ primers covering all of isotypes (reverse primers). Illumina

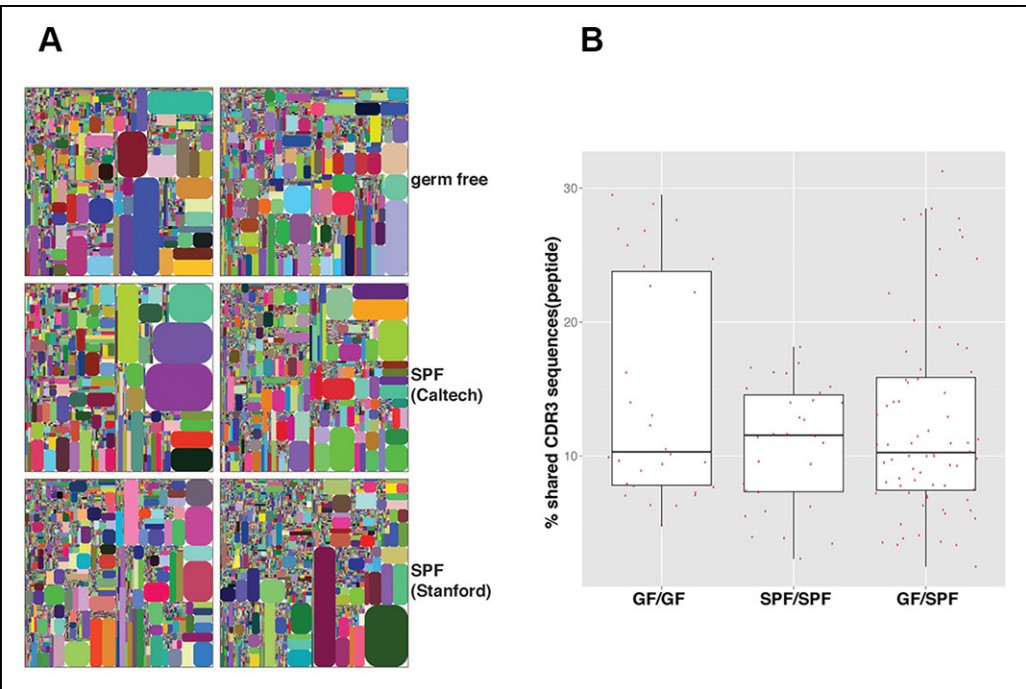

**Figure 9.** The B-1a IgH repertoires from mice raised in specific pathogen free condition are comparable to the B-1a IgH repertoire from age-matched germ-free mice. (A) IgH CDR3 tree map plots for splenic B-1a cells from GF mice (*upper panel*), or SPF mice in Caltech animal facility (*middle panel*), or SPF mice in Stanford animal facility (*bottom panel*). Each plot represents the data for a sample from a 4 month old mouse. Recurrent CDR3 (nucleotide) sequences are visualized as larger contiguously-colored rectangles in each plot. (B) CDR3 peptide pair-wise sharing analysis of IgH repertoire similarity between multiple splenic B-1a samples from age-matched GF and SPF mice. GF mice (n = 6); SPF mice (n = 6). CDR3 peptide pair-wise analysis was conducted between GF mice (GF/GF), SPF mice (SPF/SPF) and GF vs. SPF mice (GF/SPF). Each dot represents the percentage of shared CDR3 peptide sequences between two mice. There was no statistical difference between each comparison.

The following figure supplement is available for figure 9:

**Figure supplement 1.** Normal splenic B-1a compartment in GF mice.

paired-end sequencing communal primer B is linked to each forward $V_H$ primer. Illumina paired-end sequencing communal primer A and a barcode sequence of 6 nucleotides are linked to each reverse $C_H$ primers. In brief, cDNA was reverse transcribed from total RNA sample using mixture of forward $V_H$ and reverse $C_H$ primers and reagents from the OneStep RT-PCR kit (Qiagen, Valencia, CA). The first round of PCR was performed at: 50°C, 40 minutes; 95°C, 15 min; 94°C, 30 s, 58°C, 2 min, 72°C, 30 s, for 15 cycles; 94°C, 30 s, 72°C, 2 min, for 10 cycles; 72°C, 10 min. After the first round of PCR, primers were removed by Exonuclease I digestion at 37°C for 30 min (New England Biolabs, lpswich, MA). Then 2 μL of the first-round PCR products were used as templates for the second round of amplification using communal A and B primers and reagents from the Multiplex PCR kit (Qiagen). The second round PCR was performed as: 95°C, 15 min; 94°C, 30 s, 55°C, 30 s, 72°C, 30 s, for 40 cycles; 72°C, 5 min. About 400bp long PCR products were run on 2% agarose gels and purified using a gel extraction kit (Qiagen). The IgH libraries were pooled and sequenced with Illumina MiSeq pair-end read-length platform. The output of IgH sequence covers CDR2, CDR3 and the beginning of the constant region. The sequence information for all primers used for the library preparation can be found in US Patent Office (US9012148).

## Sequence analysis

Sequence reads were de-multiplexed according to barcode sequences at the 5' end of reads from the IgH constant region. Reads were then trimmed according to their base qualities with a 2-base sliding window, if either quality value in this window is lower than 20, this sequence stretches from the window to 3' end were trimmed out from the original read. Trimmed pair-end reads were joined together through overlapping alignment with a modified Needleman-Wunsch algorithm. If paired forward and reverse reads in the overlapping region were not perfectly matched, both forward and reverse reads were thrown out without further consideration. The merged reads were mapped using a Smith-Waterman algorithm to germline V, D, J and C reference sequences downloaded from the IMGT web site (*Lefranc, 2003*). To define the CDR3 region, the position of CDR3 boundaries of reference sequences from the IMGT database were migrated onto reads through mapping results and the resulting CDR3 regions were extracted and translated into amino acids.

## Artifacts removal

C57BL/6J mouse $V_H$ reference sequences were pair-wise aligned with a Smith-Waterman algorithm. Two $V_H$ reference sequences are considered related if the aligned region between them is > 200bp matched and < 6 mismatches. Two sequence reads were considered related if the best mapped $V_H$ sequences are related and the CDR3 segments have less than 1 mismatch. If two sequences are related and the frequency of the minor one is less than 5% of the dominant one, the minor one is removed from further consideration. In addition, single copy CDR3s are removed from further consideration.

To allow multiplexing of multiple samples in a single sequence run, $C_H$ primers were linked with barcodes containing 6 different nucleotides. The barcode $C_H$ primers were used in a first round RT-PCR. To compensate for potential in chemical synthetic, PCR and/or sequencing error, barcodes were designed with a Hamming distance $\geq 3$. Given that the chemical synthetic error is roughly 5% per position, there is about a 1/8000 chance that one barcode is mistakenly synthesized as another barcode. For a CDR3 with n occurrences in one sample and the same CDR3 (nucleotide sequence) with N occurrences in another sample in the same sequencing run, we calculated the probability that such a CDR3 would occur n or more times if it were due to cross-contamination, using the following formula $P = 1 - \sum_{k=1}^{n-1} \frac{e^{-\lambda} \cdot \lambda^k}{k!}$ where $\lambda$ is the expected number of errors given N reads and is computed by $\lambda = N \cdot \mu$ and $\mu$ is the cross-contamination rate which is preset as 1/8000. CDR3s that yielded p<0.001 were considered highly unlikely to be due to cross-contamination. Sequences were obtained for 60 separately sorted cell populations (details for each population are in *Table 1*).

## CDR3 tree map analysis

To draw the IgH CDR3 tree-map for each sequence sample, the entire rectangle was divided: 1st into a set of rectangles with each rectangle corresponding to a distinct $V_H$ gene segment; 2nd into a set of V-J rectangles with each rectangle corresponding to a distinct V-J; and 3rd into a set of V-J-CDR3 rectangles with each rectangle representing a distinct V-J-CDR3 combination. The rectangles are ordered based on area from largest at the bottom right to smallest at the top left. The size of an individual rectangle is proportional to the relative frequency for each V-J-CDR3 combination sequence. In order to distinguish neighboring rectangles, corners of each rectangle are rounded and each rectangles are colored randomly. Therefore, each rectangle drawn in the map represents an individual CDR3 nucleotide sequence.

## CDR3 sequence diversity (D50) measurement

D50 is a measurement of the diversity of an immune repertoire of J individuals (total number of CDR3s) composed of S distinct CDR3s in a ranked dominance configuration, where $r_1$ is the abundance of the most abundant CDR3, $r_2$ is the abundance of the second most abundant CDR3, and so on. C is the minimum number of distinct CDR3s with > = 50% total sequencing reads. D50 is given by

$$\text{Assume that } \underbrace{r_1 \geq r_2 \cdots \geq r_i \cdots \geq r_{i+1} \cdots \geq r_s}_{s} \ , \ \sum_{i=1}^{s} r_i = J$$

$$\text{if } \sum_{i=1}^{c} r_i \geq J/2 \text{ and } \sum_{i=1}^{C-1} r_i < J/2$$

$$D50 = \frac{C}{S} \times 100$$

## Mutation analysis

The forward $V_H$ primers used to amplify expressed IgH genes are located at the IgH framework region 2. To avoid primers interfering with the mutation analysis, the variable region stretching from the beginning of the CDR2 to the beginning to the CDR3 was examined for mismatches between the sequence read and the best-aligned germline reference sequence. To eliminate the impact of sequencing error on this calculation, only sequence reads with more than 4 copies were included in the mutation calculation.

## Quantification of the diversities of V(D)J recombination events for a given CDR3 peptide

For this measurement, we introduce an entropy value as the index of diversity level. Assuming a distinct CDR3 peptide sequence X in a sample is derived from n number of distinct V(D)J recombinations (nucleotide) with each frequency as $P_1$, $P_2$, ... $P_n$ respectively, the entropy for X ($E_x$) is then calculated as: $E_x = -\sum_{i=1}^{n} P_i \log_2 P_i$.

For a sample, after computing entropy values for each distinct peptide CDR3 fragments, the E values for distinct peptide CDR3 fragments are categorized into four ranges: [0, 0.5), [0.5, 1.5), [1.5, 2.5) and [2.5, $+\infty$). The higher the entropy value, the more diverse the V(D)J recombinations for a given CDR3 peptide.

## Acknowledgements

We acknowledge Megan Phillips, Jeffrey Waters, Jasmine Sosa, John Mantovani and the Stanford Shared FACS Facility for excellent assistance. We are grateful to Dr. Michel Nussenzweig (Rockefeller University) for generously providing $Aid^{-/-}$ mice. Studies are supported by the US National Institutes of Health Grants R01-AI1076434 (LAH), R01-DK078938 (SKM) and by the HudsonAlpha Institute for Biotechnology and iRepertoire (HJ).

## Additional information

### Competing interests

CW and JH: Co-founder of iRepertoire. The other authors declare that no competing interests exist.

### Funding

| Funder | Grant reference number | Author |
| --- | --- | --- |
| National Institutes of Health | R01-AI1076434 | Leonore A Herzenberg |
| National Institutes of | R01-DK078938 | Sarkis K Mazmanian |

The funders had no role in study design, data collection and interpretation, or the decision to submit the work for publication.

### Author contributions

YY, CW, JH, LAH, Conception and design, Acquisition of data, Analysis and interpretation of data, Drafting or revising the article; QY, EEBG, Acquisition of data, Analysis and interpretation of data, Drafting or revising the article; ABK, Conception and design, Analysis and interpretation of data, Drafting or revising the article; HC, Acquisition of data, Drafting or revising the article, Contributed

unpublished essential data or reagents; GQ, Acquisition of data, Analysis and interpretation of data; SKM, Analysis and interpretation of data, Drafting or revising the article, Contributed unpublished essential data or reagents

## Additional files

### Major datasets

The following dataset was generated:

| Author(s) | Year | Dataset title | Dataset URL | Database, license, and accessibility information |
|---|---|---|---|---|
| Wang C, Yang Y, Yang Q, Herzenberg LA, Han J | 2015 | C57Bl/6 mice B cell subsets IgH repertoire | http://www.ncbi.nlm.nih.gov/sra/?term=SRA440320 | Publicly available at NCBI Sequence Read Archive (accession no: SRA440320) |

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
