## [Decision Letter]

Thank you for submitting your work entitled "Distinct mechanisms define murine B cell lineage immunoglobulin heavy chain (IgH) repertoires" for peer review at *eLife*. Your submission has been favorably evaluated by Tadatsugu Taniguchi (Senior editor) and four reviewers, one of whom is a member of our Board of Reviewing Editors. One of the reviewers, Harry Schroeder, has agreed to share his identity.

The reviewers have discussed the reviews with one another and the Reviewing editor has drafted this decision to help you prepare a revised submission.

Summary:

The manuscript compares the IgH repertoires of mouse B-1a and conventional B cells from postnatal day 2 to adulthood using Hi-D FACS sorting, multiplex PCR, and deep sequencing. The data provide a detailed view of the global B-1a IgH repertoire diversity, and demonstrate its continuing generation and selection during the animal's lifetime. Remarkably, the data show that B-1a B cells, like conventional B cells, undergo AID-mediated somatic hypermutation (SHM) and Ig class switch recombination (CSR), yet, microbiota play no substantive role in either repertoire selection, SHM or CSR in B-1a B cells in contrast to conventional B cells. The manuscript thus represents a substantial contribution to B-1a B cell biology.

Essential revisions:

1) The manuscript treats B-1a B cells as originating in the neonatal spleen, but makes no effort to examine the IgH repertoire in immature B cells of the fetal liver (as distinct from bone marrow) vis-a-vis the later B-1a repertoire. Both positive and negative results from such a comparison would be informative.

2) The observation that the IgH repertoire of B-1a cells in adult mice contain a high frequency of TdT-dependent N nucleotide additions suggests that a large fraction of B-1a cells in adult mice are of postnatal and not fetal origin. However, this observation is consistent either with self-renewal of the cells that were generated during the first weeks after birth or with continuous output from the bone marrow, possibilities that cannot be discriminated based on the presented data. The interpretation in the manuscript suggesting that postnatal de novo B-1a B cell development only occurs during the first few weeks of life but not thereafter thus needs to be appropriately modified. In fact, the striking finding that B-1a cells in adult animals show a very high frequency of postnatal N nucleotide addition requires highlighting.

3) The manuscript argues that the germ-free mouse data indicate that there is an endogenous antigen-driven selection program at work, but does not rule out food antigens as sources for this selection. In the absence of data from antigen-free mice, this interpretation would best be appropriately modified as showing an absence of a role for microbiota rather than a definitive role for 'endogenous' antigens alone.

4) In the pairwise sharing analysis of IgH CDR3 sequences comparing various B cell subsets of individual mice (Figure 2), splenic B-1a cells from very young mice (day 2-7) show very high CDR3 sharing. However, since 'each dot represents the data for a sample from an individual mouse except for the day 2 splenic B-1a data, which are derived from sorted cells pooled from 8 mice', it is conceivable that the very high CDR3 sequence sharing is an artifact of comparing samples of 8 mice with each other instead of individual mice like for the other developmental time points. This needs to be clarified, with exclusion of the data points of the 2-7-day-old mice if necessary. Also, the measurement of the CDR3 sequence sharing and the exact nature of the values plotted should be better explained in the text.

5) The nomenclature of VH genes used throughout the manuscript is undefined, and needs to be clarified.

6) VH11-2/Vk9-128 and VH12-3/Vk4-91 are the two most dominant PtC-binding BCRs in the IgH repertoire of B-1a cells. The authors discuss only the VH11-2 specificity, and V(D)J rearrangements involving VH12-3 segment are not even listed among conserved CDR3 sequences (Table 2). It would be useful to comment on this and clarify if this is related to technical issues and limitations.

7) A conclusion in the Results section that '...since SHM in splenic B-1a IgVH initiates later and progresses with age, these findings suggest that peritoneal B-1a in older animals are largely derived from B-1a that have migrated from the spleen when the animals were younger' while a tenable and interesting hypothesis, belongs in the Discussion, since other scenarios remain possible.

8) While germ-free mouse data are shown all along, they are only addressed at the end. This is confusing and reorganization would be useful.

9) The tree-map plots shown in Figure 1, Figure 5 and Figure 9 highlight the repertoire biases in B-1a cells in a very intuitive way. However, with randomly assigned colors, it is impossible to trace the same specificity even across panels of the same figure – yet this is crucial for following the story. It would be advisable to add numbers to the symbols of the most highly reoccurring CDR3 sequences and correlate the numbers with the actual VH gene segments at the bottom of the figure.

10) Figure 3 requires radical revisions to reduce complexity by retaining key components of FO and B-1a B cell comparisons in the main figure and moving some components to supplementary data, rectifying invisible X-axis labels and VH gene labels. Also, the fact that the data in this figure are presented 'as the normalized distribution, which counts the value of each distinct CDR3 nucleotide sequence expressing a given VH gene as one, no matter how many of this sequence are detected' needs to be included in the text as well for the sake of clarity.

11) Figure 7 needs restructuring so that its panel order follows the text. Also, it is not clear where the graphical representation of B-1a cells from AID-null mice, mentioned in the text, is shown in this figure (Figure 7).

12) Generally, figure contents need to be comprehensively edited for readability. Similarly, the text needs proofreading as well. The Discussion could also use editing to avoid repetitions and digressions into peripheral issues such as the evolutionary origins of B-1a B cell lineage, which are not directly addressed by the present data.

---

## [Author Response]

*1) The manuscript treats B-1a B cells as originating in the neonatal spleen, but makes no effort to examine the IgH repertoire in immature B cells of the fetal liver (as distinct from bone marrow) vis-a-vis the later B-1a repertoire. Both positive and negative results from such a comparison would be informative.*

The reviewers, we believe, are referring to our statement in the manuscript Abstract and Introduction that “we track B-1a cells from their origin in neonatal spleen...” We appreciate this comment by the reviewers since the wording we used is not what we meant. We apologize for this confusion. The correct wording is “we track B-1a cells from their early appearance in neonatal spleen”.

Actually, our laboratory is working quite intensively on the origin of B-1a cells and has shown that they arise from progenitors distinct from the progenitors that give rise to B-2 cells. We are now nearing publication of a paper tracing the B-1a progenitors in the embryo to a considerably earlier time-point than neonatal spleen.

In addition, we agree with the reviewers that it would be quite informative to compare the repertoire expressed by the immature (not fully developed) B cells in the fetal liver with the repertoire of the mature B-1a in neonates and adults. However, our methods do not give us this option. That is, we are not sequencing genomic DNA, which could be recovered equally from mature and immature B cells. Instead, we are sequencing mature IgH message, which is expected to be present, if at all, only in a very small proportion of immature B cells, whether in adult bone marrow or fetal liver.

Basically, there are too few mature IgM + B cells in fetal liver for reliable IgH sequencing with current methods. In our study, we harvested late fetal liver just prior to birth (E19) and examined the B cell development by FACS. We found that the IgM + B cells (i.e., those that have completed IgH V(D)J recombination) are detectable only at very low frequencies (0.6% of CD19 + total B cells) of which only 20% express the mature B-1a CD43 + CD5 + phenotype (see FACS plots at right). The IgM + B cell frequencies are even lower in E18 fetal liver (0.2% of CD19 + B cells; data not shown). These numbers are too low for us to obtain enough material for sequencing from a feasible number of embryos.

We will be happy to include the Figure in our paper as Figure 1—figure supplement 1 if the editors wish. It responds to a question that we feel many readers may also have.

*2) The observation that the IgH repertoire of B-1a cells in adult mice contain a high frequency of TdT-dependent N nucleotide additions suggests that a large fraction of B-1a cells in adult mice are of postnatal and not fetal origin. However, this observation is consistent either with self-renewal of the cells that were generated during the first weeks after birth or with continuous output from the bone marrow, possibilities that cannot be discriminated based on the presented data. The interpretation in the manuscript suggesting that postnatal de novo B-1a B cell development only occurs during the first few weeks of life but not thereafter thus needs to be appropriately modified. In fact, the striking finding that B-1a cells in adult animals show a very high frequency of postnatal N nucleotide addition requires highlighting.*

We completely agree with the reviewers that based on the information we provided, it is hard to distinguish the possibilities between the self-renewal of B-1a cells and their de novo generation from the bone marrow. This uncertainty, however, is due to our failure to clearly state in the manuscript that the N-nucleotide distribution profile that we present is based on normalized data. We use normalized data because it specifically minimizes the impact of B-1a self-replenishment on the overall N-insertion distribution profile, and hence weights the repertoire for de novo generated sequences. Thus, we score each distinct IgH sequence containing the indicated N nucleotide insertions as a single sequence, regardless how many times this sequence was detected. We have clarified this part in the text of the Result section “The B-1a IgH repertoire integrates rearrangement from de novo B-1a development that occur mainly during the first a few weeks of life” of the manuscript to make the data more understandable.

In addition, as the reviewers requested, our modified manuscript now highlights our finding that B-1a cells in adult animals show a very high frequency of N nucleotide additions, which indicates that the majority of the B-1a cells are actually generated postnatally after Tdt is expressed.

*3) The manuscript argues that the germ-free mouse data indicate that there is an endogenous antigen-driven selection program at work, but does not rule out food antigens as sources for this selection. In the absence of data from antigen-free mice, this interpretation would best be appropriately modified as showing an absence of a role for microbiota rather than a definitive role for 'endogenous' antigens alone.*

The reviewers are correct in pointing out that the food antigens cannot be ruled out as the sources for the selection in our studies. However, we believe that self-antigens are likely to be the major driving force, if only because B-1a cells are well-known to produce anti-self antibodies. We agree that food antigens may mimic self, or vice versa. Nevertheless, Occam’s razor suggests that the minimal hypothesis, i.e., stimulation by self-antigen, rather than microbiota, is likely the major influence on the selection of the B-1a repertoire. We have revised our manuscript to reflect this point.

*4) In the pairwise sharing analysis of IgH CDR3 sequences comparing various B cell subsets of individual mice (Figure 2), splenic B-1a cells from very young mice (day 2-7) show very high CDR3 sharing. However, since 'each dot represents the data for a sample from an individual mouse except for the day 2 splenic B-1a data, which are derived from sorted cells pooled from 8 mice', it is conceivable that the very high CDR3 sequence sharing is an artifact of comparing samples of 8 mice with each other instead of individual mice like for the other developmental time points. This needs to be clarified, with exclusion of the data points of the 2-7-day-old mice if necessary. Also, the measurement of the CDR3 sequence sharing and the exact nature of the values plotted should be better explained in the text.*

In responding to the concern raised by the reviewers, we have excluded the data of the day 2 sample from the CDR3 pairwise sharing analysis. The amended Figure 2 now shows data from which the day 2 sample is excluded. In addition, as per the reviewer’s suggestion, we have now added a detailed explanation for CDR3 pair-wise sharing analysis.

*5) The nomenclature of VH genes used throughout the manuscript is undefined, and needs to be clarified.*

The nomenclature of IgH V, D, J genes used in this study is based on IMGT IgH gene nomenclature. We have made this statement in the paper to clarify this issue.

*6) VH11-2/Vk9-128 and VH12-3/Vk4-91 are the two most dominant PtC-binding BCRs in the IgH repertoire of B-1a cells. The authors discuss only the VH11-2 specificity, and V(D)J rearrangements involving VH12-3 segment are not even listed among conserved CDR3 sequences (Table 2). It would be useful to comment on this and clarify if this is related to technical issues and limitations.*

As the reviewers correctly pointed out, the sequence data we presented do not contain VH12-3 encoded sequences. The absence of these VH12-3-encoded sequences is due to a technical limitation of the methods we used. In essence, when we designed the primers for VH gene families about three years ago, we inappropriately overlooked the VH12 reference gene and only included a primer for VH12-1 gene. Since VH12-1 and VH12-3 differ in the primer-aligning region, the VH12-1 primer does not capture VH12-3-encoded sequence and hence VH12-3 encoded sequences are not included our earlier dataset, which includes the bulk of the data we report in the manuscript.

We have corrected this problem by adding VH12-3 primer in our new primer sets. While we cannot retroactively correct our initial oversight, we have now noted the problem in the text of the manuscript. We believe that this omission of VH12-3 is not fatal for the data we present. However, relevant to this issue, we have recently completed additional sequencing studies in which we compared the sequence data generated by using old version of primer set with the sequence data generated by using new version of primer set that contains the VH12-3 primer. We found that sequence results for the libraries obtained by using old and new version of primers are highly similar. In fact, the most highly represented reads from the paired data sets are identical in sequence and show similar representation order.

Specifically, we sorted two splenic B-1a populations individually from two 4 months old C57B/l6 mice. We extracted RNA from each population and divided the RNA from each into two parts. For one part, we prepared an amplified library using the old primer set and for the other, we prepared an amplified library using the new primer set. We then sequenced these amplified IgH libraries. Analysis of the resultant sequences showed that, regardless of the primer set used (old or new), the sequences obtained from the IgH library are highly similar. In essence, the top 10 highly recurring CDR3 sequences (both peptide and V(D)J recombination) are identical between each pair of libraries (shown in Figure 3—figure supplement 2). As expected, we detected VH12-3 encoded sequences from the splenic B-1a IgH libraries prepared with the new primer sets, and these VH12-3 encoded sequences included several published PtC-binding VH12-3 encode sequences, i.e., AGDYDGYWYFDV (VH12-3 D2-4 J1), AGDRDGYWYFDV (VH12-3 D3-2 J1), AGDRYGYWYFDV (VH12-3 D2-9 J1).

The mutation profiles for the libraries prepared by using either old or new primer sets are also highly similar. In two separate comparisons, we detected identical IgH sequences with identical nucleotides substitutions in each library (see Figure 7—figure supplement 3).

We hope the editor will agree that while there is a clear need to collect VH12-3 data in future studies, the absence of this sequence in our current data is not a fatal flaw. To resolve this issue for the journal readers, we would be pleased to include the comparison data presented here in the supplement data of the manuscript. Importantly, these comparisons also show the reliability of the technology we have developed for this study.

*7) A conclusion in the Results section that '...since SHM in splenic B-1a IgVH initiates later and progresses with age, these findings suggest that peritoneal B-1a in older animals are largely derived from B-1a that have migrated from the spleen when the animals were younger' while a tenable and interesting hypothesis, belongs in the Discussion, since other scenarios remain possible.*

We agree with reviewers and have moved this part to the Discussion section.

*8) While germ-free mouse data are shown all along, they are only addressed at the end. This is confusing and reorganization would be useful.*

We understand the reviewer’s concern about our decision to show the germ-free data throughout the Results section but to put off discussion of these germ-free data until the last section of the Results, in which we explicitly focus on addressing the germ-free data. We did this to efficiently utilize the space available in the manuscript. If the editor wants to relax this restriction, we will be please to discuss this material in the text when it appears. As an alternate solution, we have added the statement “The data for germ-free animals is discussed at the end of the Results section” to the current table and figure legends.

*9) The tree-map plots shown in Figure 1, Figure 5 and Figure 9 highlight the repertoire biases in B-1a cells in a very intuitive way. However, with randomly assigned colors, it is impossible to trace the same specificity even across panels of the same figure – yet this is crucial for following the story. It would be advisable to add numbers to the symbols of the most highly reoccurring CDR3 sequences and correlate the numbers with the actual VH gene segments at the bottom of the figure.*

Following the reviewers’ suggestion, we have revised Figure 1 by adding numbers to identify the top four highly recurring CDR3 sequences that are shared in splenic B-1a and peritoneal B-1a CDR3 tree-map plots. For Figure 5 and Figure 9 which there are many tree-map plots showing recurring CDR3 sequences, we added a supplementary table in which we list the top ten CDR3 sequences for the CDR3 tree-map plot that show recurring CDR3 sequences in each figure.

*10) Figure 3 requires radical revisions to reduce complexity by retaining key components of FO and B-1a B cell comparisons in the main figure and moving some components to supplementary data, rectifying invisible X-axis labels and VH gene labels. Also, the fact that the data in this figure are presented 'as the normalized distribution, which counts the value of each distinct CDR3 nucleotide sequence expressing a given VH gene as one, no matter how many of this sequence are detected' needs to be included in the text as well for the sake of clarity.*

We have re-done Figure 3 following the reviewers’ instruction. To better match the text, we show the comparison between MZB and splenic B-1a cells from 2-6 months old mice. In addition, as reviewers requested, we also now include the requisite statement in the text as well as in the figure legend.

*11) Figure 7 needs restructuring so that its panel order follows the text. Also, it is not clear where the graphical representation of B-1a cells from AID-null mice, mentioned in the text, is shown in this figure (Figure 7).*

Following the reviewers’ advice, the panels in Figure 7 have been re-organized to correspond more closely to the text.

In Figure 7 have shown data for each B cell sample obtained from 4-5 months old AID knockout mice. As we state in the Figure 7 legend, there are a total of seven B cell samples for which data are shown. These include four splenic B-1a samples, each from an individual mouse; one peritoneal B-1a sample; one follicular B and one peritoneal B-2 samples. We have clarified the text to match.

12) Generally, figure contents need to be comprehensively edited for readability. Similarly, the text needs proofreading as well. The Discussion could also use editing to avoid repetitions and digressions into peripheral issues such as the evolutionary origins of B-1a B cell lineage, which are not directly addressed by the present data.

Following the reviewers’ suggestion, we have comprehensively edited our figure contents and text to make them clearer and more understandable. We also removed the repetitions in the Discussion section. Our sequence findings are consistent with developmental data that we and other labs have presented showing that B-1a and B-2 cells belong to evolutionarily distinct B lineages. While we agree with reviewers that our data do not directly bear on this lineage distinction, we believe that the drastic differences between the repertoire-defining mechanisms in B-1a and B-2 is consistent with, and sheds light on, the B-1a/B-2 lineage split. We agree with the reviewers that the “lineage” discussion in our manuscript should be shortened and have modified the text of the Discussion in the manuscript.